# PlioMIP2 simulations with NorESM-L and NorESM1-F

Xiangyu Li[1, 2], Chuncheng Guo[2], Zhongshi Zhang[3, 2, 4, *], Odd Helge Otterå[2], Ran Zhang[1]

[1]Climate Change Research Center, Institute of Atmospheric Physics, Chinese Academy of Sciences, Beijing 100029, China

[2]NORCE Norwegian Research Centre, Bjerknes Centre for Climate Research, Bergen, Norway

[3]Department of Atmospheric Science, School of Environmental Studies, China University of Geosciences, Wuhan, China

[4]Nansen-Zhu International Research Centre, Institute of Atmospheric Physics, Chinese Academy of Sciences, Beijing 100029, China

*Correspondence to:* Zhongshi Zhang, E-mail address: zhzh@norceresearch.no

**Abstract.** As a continuation of the Pliocene Model Intercomparison Project (PlioMIP), PlioMIP Phase 2 (PlioMIP2) coordinates a wide selection of different climate model experiments aimed at further improving our understanding of the climate and environments during the late Pliocene with updated boundary conditions. Here we report on PlioMIP2 simulations carried out by the two versions of the Norwegian Earth System Model (NorESM), NorESM-L and NorESM1-F, with updated boundary conditions derived from the Pliocene Research, Interpretation and Synoptic Mapping version 4 (PRISM4). NorESM1-M is the version of NorESM that contributed to the Coupled Model Intercomparison Project Phase 5 (CMIP5). NorESM-L is the low-resolution of NorESM1-M, whereas NorESM1-F is a computationally efficient version of NorESM1-M, with similar resolutions and updated physics. Relative to NorESM1-M, there are notable improvements in simulating the strength of the AMOC and the distribution of sea ice in NorESM1-F, partly due to the updated ocean physics. The two NorESM versions both produce warmer and wetter Pliocene climate, with a greater warming over land than over ocean. Relative to the pre-industrial period, the simulated Pliocene global mean surface air temperature is 2.1°C higher with NorESM-L and 1.7°C higher with NorESM1-F, respectively, and the corresponding global mean sea surface temperature enhances by 1.5°C and 1.2°C. The simulated precipitation for the Pliocene increases by 0.14 mm day$^{-1}$ globally in both model versions, with large increases in the tropics and especially in the monsoon regions and only minor changes, or even slight

decreases, in subtropical regions. The intertropical convergence zone (ITCZ) shifts northward in the Atlantic and Africa in boreal summer. In the simulated warmer and wetter Pliocene world, Atlantic meridional overturning circulation (AMOC) becomes deeper and stronger, with the maximum AMOC levels increasing by ~9% (with NorESM-L) and ~15% (with NorESM1-F), while the meridional overturning circulation slightly strengthens in the Pacific and Indian Oceans. Although the two models produce similar Pliocene climates, they also generate some differences, in particular for the Southern Ocean and the northern middle and high latitudes, which should be investigated through the PlioMIP2 in the future. As compared to PlioMIP1, the simulated Pliocene warming with NorESM-L is weaker in PlioMIP2, but otherwise show very similar responses.

## 1 Introduction

The mid-Pliocene warm period (mPWP, 3.0~3.3 million years ago), a warm and stable interval in the Earth's geological history with paleogeography configurations and greenhouse gas concentrations similar to today, provides an interesting case study for understanding possible warm climates in our future. During the mPWP, the estimated global mean temperature was about 2–3 ℃ higher than at present (e.g., Dowsett et al., 2009, 2010a). The global mean sea level was higher than that of today, with a peak of 22±10 meters (Miller et al., 2012). Due to warm sea surface temperatures, the Arctic Ocean experienced seasonally sea ice free conditions (Cronin et al., 1993; Dowsett et al., 2010b; Robinson, 2009; Clotten et al., 2018). Ice sheets were smaller over western Antarctica and Greenland and the vegetation belts were displaced poleward (Salzmann et al., 2008; Dowsett et al., 2010b).

The warm mPWP climate has been simulated with a suite of models under the framework of Pliocene Model Intercomparison Project phase 1 (PlioMIP1), forced with the boundary conditions from PRISM version 3 (PRISM3) (Dowsett et al., 2010a; Haywood et al., 2013). According to these simulations, the simulated global annual mean surface temperature during the mPWP was 1.8–3.6 ℃ above pre-industrial levels (Haywood et al., 2010, 2013, 2016a). The high atmospheric $CO_2$ level (405ppmv) dominated the majority of warming in tropical regions, while clear-sky albedo was mainly responsible for a stronger warming at high latitudes (Hill et al. 2014). Furthermore, the Hadley Cell became weaker and shifted poleward (Sun et al., 2013). Westerlies (Li et al., 2015) and global tropical cyclones (Yan et al., 2016) migrated poleward. Also, according to the models, the East Asian summer monsoon intensified in the Pliocene warm climate (Zhang, R. et al., 2013), and the global monsoon

system generally brought more precipitation into the expanded monsoon regions (Li et al., 2018). The simulated Arctic sea ice was less extensive and thinner than it is in modern times (Howell et al., 2016). The simulated Atlantic meridional overturning circulation (AMOC) and the associated ocean heat transport were similar to those of the pre-industrial period (Zhang, Z. et al., 2013a). Despite the generally consistent features of the simulations, a large model-data mismatch in terms of the warming magnitude remained at the northern high latitudes (e.g., Atlantic, Arctic Ocean, and Asia) (Dowsett et al., 2013; Haywood et al., 2013).

To better understand the warm mPWP climate and to better constrain the model-data mismatch, PlioMIP phase 2 (PlioMIP2) was launched in 2016. The modelling strategy adopted in PlioMIP2 has been revised to establish a more well-balanced methodology for model-data comparisons. Instead of focusing on the time window between 3.0 and 3.3 Ma, PlioMIP2 identified the time slice of 3.205 Ma centered on an interglacial peak (marine isotope stage KM5c) as the key target for the model-data comparison (Haywood et al., 2016b). The updated boundary conditions taken from PRISM4 (Dowsett et al., 2013; Salzmann et al., 2013; Haywood et al., 2016b) are used in PlioMIP2.

In this study, we present PlioMIP2 simulations with two different versions of the NorESM, NorESM-L (Zhang et al., 2012) and NorESM1-F (Guo et al., 2019). In the following sections, we first introduce the two model versions and our experimental design. Next, we present the simulated results in section 4 and discussions in section 5. Finally, the last section gives the summary and conclusions.

## 2 Model descriptions

The NorESM is developed based on the structure of the Community Climate System Model version 4 (CCSM4) from the National Centre for Atmospheric Research (Gent et al., 2011). In the model, the atmospheric component is the Oslo version of CAM4 (CAM4-Oslo), which implements an advanced scheme for interactions between aerosol and clouds (Kirkevåg et al., 2013). The oceanic component is the Miami Isopycnic Coordinate Ocean Model (MICOM) (Bleck and Smith, 1990; Bleck et al., 1992; Bentsen et al., 2013), with several improvements. To limit model complexity and speed up model integration, both NorESM-L and NorESM1-F use the standard, prescribed aerosol chemistry of CAM4 rather than that of CAM4-Oslo.

### 2.1 NorESM-L

NorESM-L is a low-resolution of NorESM1-M (the version of NorESM that contributed to the Coupled Model Intercomparison Project Phase 5, CMIP5), and is developed for paleoclimate simulations (Zhang et al., 2012). The atmospheric component has a horizontal resolution of T31 (~3.75°) and 26 vertical levels. The ocean component employs a bipolar grid with a horizontal resolution of nominal 3°, and uses 30 isopycnic vertical layers (Table 1). NorESM-L was used to simulate the Pliocene climate in PlioMIP1 (Zhang et al., 2012). Further information on NorESM-L is documented in Zhang et al. (2012).

## 2.2 NorESM1-F

NorESM1-F is assembled for long time simulations with relatively high resolutions and improved process representations and climate performance compared to NorESM1-M (Guo et al., 2019). In the model, the atmosphere component uses a horizontal resolution of 1.9° latitude and 2.5° longitude and uses 26 vertical levels. The ocean component employs a tripolar grid with a nominal 1° horizontal resolution and uses 53 vertical layers (Table 1).

In NorESM1-F, the change of ocean-sea ice grid from bipolar in NorESM1-M to tripolar configuration, together with a reduction of model complexity by replacing the comprehensive aerosol–cloud process representations in NorESM1-M with the standard prescribed aerosol chemistry of CAM4 (as was done in NorESM-L), lead to large speedup of the model. In addition, the reduction in the coupling frequency between atmosphere–sea ice and atmosphere–land and the dynamic subcycling of the sea ice is helpful to improve the computational performance with a relatively small effect on the modelled climate (Guo et al., 2019).

Compared to NorESM1-M, there are some updates in the ocean physics in NorESM1-F. NorESM1-F takes measures to reduce sea ice thickness biases in shelf regions and modifies the parameterization of oceanic mesoscale eddies and the vertical mixing (Guo et al., 2019). With those updates to the ocean physics, NorESM1-F provides reasonable simulations of sea ice and AMOC compared to NorESM1-M (Guo et al., 2019). In the atmosphere component, the model adopts the formulation for energy updates and energy conservation, changes the air-sea flux calculation, and modifies the calculation of the solar zenith angle (Guo et al., 2019). Compared to NorESM1-M, the seasonal cycle of sea surface temperature in the equatorial Pacific is markedly improved with NorESM1-F. There are also several important improvements on how precipitation is simulated, e.g.,

improvements in seasonality, reduced wet bias, and mitigation of the common double-ITCZ problem. Further details on the model performance of NorESM1-F can be found in Guo et al. (2019).

## 3 Experimental designs

### 3.1 Pre-industrial control experiment

According to the PlioMIP2 protocol (Haywood et al., 2016b), we use modern geographic boundary conditions, including modern land-sea mask, topography, and ice sheets and vegetation for year 1850, in the pre-industrial control experiments (Table 2). For NorESM-L, we set atmospheric $CO_2$, $N_2O$, and $CH_4$ levels to the pre-industrial values of 280 ppmv, 270 ppbv, and 760 ppbv, respectively. The orbital parameters apply values for year 1950. For NorESM1-F, the default pre-industrial

atmospheric $CO_2$, $N_2O$, and $CH_4$ levels are 284.7 ppmv, 275.68 ppbv, and 791.6 ppbv, respectively.

The pre-industrial experiment with NorESM-L was run for 2200 years, and the experiment with NorESM1-F was run for 2000 years. Climatological means of the last 100 years were analyzed in this study.

### 3.2 Pliocene experiment

Following PlioMIP2 experimental guidelines (Haywood et al., 2016b), we close the Hudson Bay, Bering Strait, and straits through the Canadian Arctic Archipelago in the Pliocene land-sea configuration. The atmospheric $CO_2$ concentration is set to 400 ppmv. Atmospheric $N_2O$ and $CH_4$ concentrations, the solar constant, and orbital parameters are identical to pre-industrial values (Table 2).

We use the "anomaly method" recommended in PlioMIP2 to create the paleogeography for the

Pliocene experiment. We first calculate differences between the PRISM4 Pliocene and PRISM4 modern topography and interpolate these to a T31 resolution for NorESM-L and to a 1.9° x 2.5° resolution for NorESM1-F. Then, we add the interpolated topography anomalies to modern topography in the pre-industrial experiment.

To create vegetation in the Pliocene experiment, we first interpolate PRISM4 Pliocene vegetation

to the resolution for NorESM-L and for NorESM1-F. Then, we convert biome vegetation types to LSM (Land System Model) vegetation types following the procedure outlined by Rosenbloom (2009). Lakes for the Pliocene are prescribed by adding the PRISM4 lake area anomaly to modern conditions. Pliocene soil conditions remain the same as the pre-industrial conditions.

With NorESM-L, the Pliocene experiment was run for 1200 years. With NorESM1-F, the Pliocene experiment was first spun up for 2000 years with atmospheric $CO_2$ concentration set to 400 ppmv, and then run for 500 years forced with all Pliocene boundary conditions. Climatological means of the last 100 years were used for analysis.

## 4 Results

### 4.1 Surface air temperature

Relative to the pre-industrial experiments, the simulated Pliocene climate is warmer according to both NorESM versions (Fig. 1). The global annual mean surface air temperature (SAT) increases by 1.7°C and 2.1°C under the NorESM1-F and NorESM-L Pliocene simulations, respectively. In particular, stronger warming appears at high latitudes (Fig. 1 and Table 3). The simulated Pliocene annual mean SAT increases by 5.2°C (NorESM1-F) and 4.9°C (NorESM-L) at the northern high latitudes and by 3.2°C (NorESM1-F) and 7.6°C (NorESM-L) at the southern high latitudes. Weak cooling appears in tropical Africa, India, northeastern Asia, northern Australia, and southern Pacific close to western Antarctica, under the NorESM1-F and NorESM-L Pliocene simulations (Fig. 1a, b). Both NorESM1-F and NorESM-L simulate stronger warming over land than over ocean. Relative to the pre-industrial period, the simulated Pliocene global mean surface air temperature (SAT) over land increases by 2.3°C with NorESM-L and 2.0°C with NorESM1-F, which is notably larger than the warming over ocean (2.0°C and 1.6°C for the NorESM-L and the NorESM1-F, respectively). This stronger warming over land is a common feature in most PlioMIP2 simulations. However, the simulated zonal mean SAT over land is nearly twice as large as in the ocean at the northern high latitudes (Fig. S1).

Changes in seasonal SAT follow a similar pattern to those of the annual SAT. The two models generate strong seasonal warming in Circum-Arctic regions, e.g., Hudson Bay, the Canadian Arctic Archipelago, and Greenland in the Pliocene experiment. However, NorESM-L produces larger Pliocene seasonal warming over the Southern Ocean and Antarctica than NorESM1-F.

### 4.2 Precipitation

The simulated Pliocene global mean annual precipitation increases by 0.14 mm day$^{-1}$ according to the two NorESM versions (Fig. 2). The mean annual precipitation increases largely in the tropical regions and especially in monsoon regions of North Africa, Asia, and Australia. In subtropical regions,

the zonal mean annual precipitation does not change markedly or slightly decreases. The changes in seasonal precipitation generally follow the pattern of annual precipitation. Both models suggest that the ITCZ shifts northward in the Atlantic and in Africa in the boreal summer, but does not change considerably in the boreal winter.

**4.3 Sea surface temperature**

Both NorESM1-F and NorESM-L simulate higher sea surface temperatures (SSTs) in the Pliocene experiments compared to pre-industrial experiments (Fig. 3). The simulated Pliocene global mean annual SST is 1.2°C (NorESM1-F) and 1.5°C (NorESM-L) higher than pre-industrial level (Fig. 3 and Table 3). Large increases in SST appear at the high latitudes of both hemispheres (Fig. 3 and Table 3). The simulated Pliocene annual mean SST increases by 2.4°C (NorESM1-F) and 2.1°C (NorESM-L) in the middle and high latitudes of the North Atlantic (north of 30 °N) and by 1.4°C (NorESM1-F) and 2.4°C (NorESM-L) in the Southern Ocean. A slight cooling occurs in the Southern Pacific close to western Antarctica according to the Pliocene experiments for both NorESM model versions. The seasonal change in SST is largely consistent with the annual pattern.

**4.4 Sea surface salinity**

Changes in sea surface salinity (SSS) from the Pliocene experiments differ notably between NorESM1-F and NorESM-L. The global mean SSS decreases by 0.34 g kg$^{-1}$ in the NorESM1-F Pliocene simulation, while it increases by 0.16 g kg$^{-1}$ according to the NorESM-L Pliocene simulation (Fig. 4). With NorESM1-F, Pliocene SSS generally decreases in most oceans, except for Baffin Bay, the Labrador Sea, and the North Atlantic subpolar regions. With NorESM-L, Pliocene SSS slightly increases in most oceans while it decreases in the Indian Ocean and in the Arctic.

The divergent responses in SSS in NorESM1-F and NorESM-L are most likely to be associated with the different vertical redistribution of salt in the two models, due to differences in e.g. surface layer mixing, ocean ventilation, convection and circulation. The two models have different vertical resolutions and horizontal/vertical mixing schemes. When the drift in global mean SSS is removed, NorESM-L and NorESM1-F show similar regional anomalies (Fig. S2). Both versions show that the SSS contrast among the Indian Ocean, the Arctic and the rest of the oceans is intensified in the Pliocene experiment (Fig. S2).

### 4.5 Sea ice

The simulated Pliocene sea ice with NorESM1-F and NorESM-L is reduced both in terms of its thickness and its extent (Fig. 5). In the Northern Hemisphere, the simulated Pliocene sea ice thickness in the Arctic Ocean is reduced by 0.5 to 1 m in March, and by more than 1 m in September. In September, NorESM1-F simulates an almost ice-free Arctic in the Pliocene experiment, while sea ice remains in the central Arctic according to the NorESM-L Pliocene simulation. In March, sea ice becomes thinner while still covering most of the Arctic Ocean in both models. In the Southern Hemisphere, although both models generate retreated sea ice extents for the Southern Ocean, NorESM-L simulates the larger sea ice responses. The Southern Ocean becomes almost ice-free specifically in March according to the NorESM-L Pliocene simulation.

### 4.6 Meridional overturning circulation

The simulated Pliocene AMOC becomes stronger and deeper in both models compared to the pre-industrial climate. With NorESM1-F, the maximum AMOC is 28.1 Sv in the Pliocene experiment, increasing by about 15% (Fig. 6 and Table 4). With NorESM-L, the simulated maximum AMOC is 23.3 Sv in the Pliocene experiment, which is 2 Sv (about 9%) larger than that in the pre-industrial experiment. Both models suggest that the vertical extent of the AMOC cell penetrates deeper during the Pliocene relative to the pre-industrial period (Fig. 6 and Table 4). Compared to the pre-industrial period, Pliocene SSS increases in Baffin Bay, the Labrador Sea, and the North Atlantic subpolar gyre in both models (Fig. 4), reducing the surface stratification and tending to favor more open ocean convection, thereby potentially contributing to the strengthened AMOC.

In the Pacific and Indian Ocean, meridional overturning circulation is slightly stronger in the Pliocene experiments than that in the pre-industrial experiments. As for the shallower circulation in the Northern Pacific subtropical gyre, the maximum of the circulation is increased by 1.0 Sv with NorESM1-F (22.3 Sv for the pre-industrial experiment *vs.* 23.3 Sv for the Pliocene experiment) and by 0.9 Sv with NorESM-L (30.7 Sv for the pre-industrial experiment *vs.* 31.6 Sv for the Pliocene experiment) (Fig. 6 and Table 4). As for the deeper circulation associated with Pacific Deep Water, the strength is increased by 0.8 Sv with NorESM1-F (–13.6 Sv for the pre-industrial experiment *vs.* –14.6 Sv for the Pliocene experiment) and by 4.6 Sv with NorESM-L (–17.0 Sv for the pre-industrial experiment *vs.* –21.6 for the Pliocene experiment) (Fig. 6 and Table 4). In the Pliocene experiments,

both NorESM versions simulate an extended northward penetration of deep water.

## 5 Discussions

### 5.1 NorESM1-F vs. NorESM1-L

Although the two versions of the NorESM simulate similar Pliocene climates, they still exhibit some differences. The most significant differences appear in the Southern Ocean (Fig. 1). SST increase over the Southern Ocean is ~1°C larger with NorESM-L than with NorESM1-F in the Pliocene experiments. This difference between the two model versions is likely to be associated with different responses in ocean heat transport and sea ice. Simulated Pliocene southward ocean heat transport to the Southern Ocean is reduced according to NorESM1-F, but increased according to NorESM-L (Fig. 7a and 7c), which partly explains the reduction in the Southern Ocean sea ice extent being more pronounced for NorESM-L than it is according to NorESM1-F (Fig. 5). Pliocene (austral summer) sea ice is nearly absent according to NorESM-L, while it still covers part of the Southern Ocean according to NorESM1-F. On the one hand, the larger seasonal warming in the Southern Ocean favors less sea ice extent in the Pliocene experiment simulated with NorESM-L. On the other hand, the presence of less sea ice leads to a reduction in albedo and to a more active ocean-atmosphere interaction (e.g. ice-albedo feedback), and contributes to the higher levels of Southern Ocean warming in the Pliocene experiment simulated with NorESM-L.

NorESM-L also simulates increased ventilation in the Southern Ocean, while NorESM1-F does not. As is indicated by the changes in salinity, sea water over the Southern Ocean becomes less stratified according to the NorESM-L Pliocene simulation (Fig. 8). The weakened ocean stratification allows the Southern Ocean to be well ventilated. As a result, the simulated Pliocene deep water is much younger in the Southern Ocean (Fig. S3). This well-ventilated Southern Ocean also appears from the PlioMIP1 Pliocene simulation with NorESM-L (Zhang, Z. et al., 2013b). However, with NorESM1-F, simulated Pliocene Southern Ocean stratification appears similar to that simulated in the pre-industrial experiment. Such divergent responses in Southern Ocean stratifications also appeared in the PlioMIP1 simulations (Zhang, Z. et al., 2013a). It remains difficult to fully explain the divergent responses. The explanation is likely related to the updated ocean physics and/or higher resolution in NorESM1-F, when compared to NorESM-L.

The other remarkable differences in the Pliocene simulations with the two versions appear in the northern middle and high latitudes. Compared to NorESM-L, the increase in the Pliocene SST simulated with NorEMS1-F at the northern middle (high) latitudes is 0.2°C (0.1°C) larger (Fig. 3 and Table 3). The increase in the Pliocene annual mean SAT at the northern middle (high) latitudes with NorESM1-F is 0.2°C (0.3°C) larger (Fig. 1 and Table 3). The stronger Pliocene warming at the northern high latitudes is most likely related to the mechanism responsible for the larger responses in sea ice reduction with NorESM1-F, since the clear sky albedo, particularly in sea ice regions, dominates the high latitudes warming in Pliocene (Hill et al., 2014). In associated with the larger salinity increase in the northern North Atlantic (Fig. 4), the enhancement of AMOC is larger with NorESM1-F than with NorESM-L (~15% *vs*. ~9%), which favors the larger increases in the Pliocene northward ocean heat transport to the Atlantic with NorESM1-F (Figs. 6 and 7). Correspondingly, the less sea ice simulated in the Pliocene experiment contributes to a larger warming at the high latitudes with NorESM1-F than with NorESM-L through the ice-albedo feedback (Figs. 1 and 3).

**5.2 PlioMIP1 vs. PlioMIP2**

With NorESM-L, the simulated Pliocene surface temperatures in PlioMIP2 are slightly cooler than those simulated in PlioMIP1. Simulated increases in global annual mean SAT and SST are 1.1°C and 0.43°C less in PlioMIP2 than those generated in PlioMIP1 (Fig. 9). At the northern middle and high latitudes, the simulated increases in annual mean SAT (SST) are 1.7°C and 5.0°C (1.0°C and 0.7°C) weaker, respectively, in PlioMIP2 than in PlioMIP1 (Fig. 9, and Table 5). Simulated weaker levels of Pliocene warming also appear in the PlioMIP2 experiment with HadCM3, according to which the Pliocene annual mean SAT is 0.4°C cooler in PlioMIP2 (Hunter et al., 2019) than in PlioMIP1 (Bragg et al., 2012). However, MRI-CGCM2.3, CCSM4, and IPSL-CM5A all simulate a larger warming in the Pliocene experiments in PlioMIP2 (Kamae et al., 2016; Chandan et al., 2017; Tan et al., 2019) than those in PlioMIP1 (Contoux et al. 2012; Kamae and Ueda, 2012; Rosenbloom et al., 2013).

The simulated weaker Pliocene warming is attributable to the modified boundary conditions used in PlioMIP2. Relative to PlioMIP1, the Pliocene atmospheric $CO_2$ is less by 5 ppmv in PlioMIP2 (400 ppmv in PlioMIP2 *vs*. 405 ppmv in PlioMIP1). The slight reduction of Pliocene atmospheric $CO_2$ is not likely to induce significant effect on the simulated Pliocene warming weakness. Compared to PlioMIP1, the seaways at the northern high latitudes (including the Bering Strait and Canadian Artic Archipelago

straits) are closed under the PlioMIP2 boundary conditions. Previous sensitivity investigations have demonstrated that the closure of the Bering Strait favors the formation of North Atlantic deep water and the intensification of AMOC by reducing freshwater transport from the Pacific to the Arctic Ocean and the North Atlantic (e.g., Hu et al., 2010; Brierley and Fedorov, 2016; Otto-Bliesner et al., 2017). In contrast, the closure of Canadian Arctic Archipelago Straits tends to weaken AMOC, by increasing fresh water transport through the Fram Strait and thereby leading to the freshening and subsequent cooling of the Labrador and Greenland-Iceland-Norwegian Seas (Otto-Bliesner et al., 2017). However, the combined effects of closing these two seaways are complicated and appear to be model dependent. For example, with the Bering Strait and Canadian Arctic Archipelago Straits closed together, Pliocene ocean surface warming in the North Atlantic high latitudes is enhanced under the CCSM4 (Otto-Bliesner et al., 2017) and IPSL-CM5A simulations (Tan et al., 2019), but is weakened according to NorESM-L.

**6 Conclusions**

In this study, we used two versions of the NorESM (NorESM-L and NorESM1-F) to carry out core experiments designed in PlioMIP2, with boundary conditions derived from PRISM4. Relative to the pre-industrial period, the simulated Pliocene global mean SAT is 2.1℃ higher according to NorESM-L and 1.7℃ higher according to NorESM1-F. The simulated Pliocene global mean SST is 1.5℃ warmer according to NorESM-L and 1.2℃ warmer according to NorESM1-F. Compared to NorESM1-F, the simulated Pliocene warming is larger with NorESM-L. In both model versions, Pliocene global mean precipitation increases by 0.14 mm/day. Strong precipitation responses occur in tropical regions and especially in monsoon regions, and, the ITCZ shifts northward in the Atlantic and in Africa in the boreal summer. According to the Pliocene experiments of both NorESM versions, AMOC becomes stronger and deeper, and meridional overturning circulation strengthens slightly in the Pacific and Indian Oceans. Although the two models simulate similarly warm climates for the Pliocene, they also produce some differences and especially for the Southern Ocean and the northern middle and high latitudes. NorESM-L simulates increased ventilation in the Pliocene Southern Ocean, while NorESM1-F does not. Compared to the climate simulated with NorESM-L in PlioMIP1, the Pliocene warming simulated in PlioMIP2 (with the updated PRISM4 boundary conditions) is slightly less pronounced. A comparison of Pliocene climates simulated with NorESM-L in PlioMIP1 and PlioMIP2

shows a weaker warming in PlioMIP2. The Pliocene global mean SAT and SST are 1.1℃ and 0.43℃ lower, respectively, in PlioMIP2 than those in PlioMIP1. Sensitivity experiments testing the impacts of boundary conditions modification made from PlioMIP1 to PlioMIP2, i.e., the effects of closing ocean gateways in the northern high latitudes, will be helpful in casting further light on these model

discrepancies. The model–dependent sensitivity to the closure of the ocean gateways in the northern high latitudes will be an interesting question that is worth further attention within the PlioMIP2 community.

*Data availability*. All PlioMIP2 boundary conditions are available on the USGS PlioMIP2 web page (https://geology.er.usgs.gov/egpsc/prism/7_pliomip2.html). Climatological averages of the two

NorESM versions will be uploaded to the PlioMIP2 data repository later (sftp://see-gw-01.leeds.ac.uk). Request of access should be directed to A. M. Haywood. Specific data can be obtained upon requests to the corresponding author Zhongshi Zhang (zhzh@norceresearch.no).

*Author Contributions*: Z. Z. and X. L. designed the study and developed the structure of this work. Z. Z. conducted the NorESM-L simulations. C. G. conducted the NorESM1-F simulations. X. L. analyzed all

the data, wrote the manuscript. All co-authors contributed to writing the manuscript and discuss the data analysis. Correspondence and requests for materials should be addressed to Z. Z.

*Competing interests*: The authors declare that they have no conflict of interest.

*Acknowledgements*: This work was supported by the National Key Research and Development Program of China (2018YFA0605602), the National Natural Science Foundation of China (41888101, 41472160,

41775088), the Norwegian Research Council (Project No. 221712, 229819), and the Thousand Talents Program for Distinguished Young Scholars (Zhongshi Zhang). X. L. gratefully acknowledges financial support from China Scholarship Council (201804910023) and the China Postdoctoral Science Foundation funded project (2015M581154). The NorESM simulations benefitted from resources provided by UNINETT Sigma2 – the National Infrastructure for High Performance Computing and

Data Storage in Norway. We thank Dr. Baohuang Su for valuable discussion. We further thank the reviewers and editor for constructive suggestions and improvement to the manuscript.

**Referrences**

Bentsen, M., Bethke, I., Debernard, J.B., Iversen, T., Kirkevåg, A., Seland, Ø., Drange, H., Roelandt, C., Seierstad, I.A., Hoose, C., and Kristjánsson, J.E.: The Norwegian Earth System Model, NorESM1-M – Part 1: Description and basic evaluation of the physical climate. Geosci. Model Dev. 6, 687–720, 2013.

Bleck, R., Rooth, C., Hu, D., and Smith, L.T.: Salinity-driven Thermocline Transients in a Wind- and Thermohaline-forced Isopycnic Coordinate Model of the North Atlantic. J. Phys. Oceanogr., 22, 1486–1505, 1992.

Bleck, R., and Smith, L.T.: A wind-driven isopycnic coordinate model of the north and equatorial Atlantic Ocean: 1. Model development and supporting experiments. J. Geophys. Res., 95, 3273–3285, 1990.

Braconnot, P., Otto-Bliesner, B., Harrison, S., Joussaume, S., Peterchmitt, J.Y., Abe-Ouchi, A., Crucifix, M., Driesschaert, E., Fichefet, T., Hewitt, C.D., Kageyama, M., Kitoh, A., Loutre, M.F., Marti, O., Merkel, U., Ramstein, G., Valdes, P., Weber, L., Yu, Y., and Zhao, Y.: Results of PMIP2 coupled simulations of the Mid-Holocene and Last Glacial Maximum – Part 2: feedbacks with emphasis on the location of the ITCZ and mid- and high latitudes heat budget. Clim. Past 3, 279–296, 2007.

Bragg, F.J., Lunt, D.J., and Haywood, A.M.:. Mid-Pliocene climate modelled using the UK Hadley Centre Model: PlioMIP Experiments 1 and 2. Geosci. Model Dev. 5, 1109–1125, 2012.

Brierley, C.M., and Fedorov, A.V.: Comparing the impacts of Miocene–Pliocene changes in inter-ocean gateways on climate: Central American Seaway, Bering Strait, and Indonesia. Earth Planet. Sci. Lett. 444, 116–130, 2016.

Chandan, D., and Peltier, W.R.: Regional and global climate for the mid-Pliocene using the University of Toronto version of CCSM4 and PlioMIP2 boundary conditions. Clim. Past 13, 919–942, 2017.

Clotten, C., Stein, R., Fahl, K., and De Schepper, S.: Seasonal sea ice cover during the warm Pliocene: Evidence from the Iceland Sea (ODP Site 907). Earth Planet. Sci. Lett. 481, 61–72, 2018.

Contoux, C., Ramstein, G., and Jost, A.: Modelling the mid-Pliocene Warm Period climate with the IPSL coupled model and its atmospheric component LMDZ5A. Geosci. Model Dev. 5, 903–917,

2012.

Cronin, T.M., Whatley, R., Wood, A., Tsukagoshi, A., Ikeya, N., Brouwers, E.M., and Briggs, W.M.: Microfaunal evidence for elevated Pliocene temperatures in the Arctic Ocean. Paleoceanography 8, 161–173, 1993.

Dowsett, H.J., Robinson, M.M., and Foley, K.M.: Pliocene three-dimensional global ocean temperature reconstruction. Clim. Past 5, 769–783, 2009.

Dowsett, H., Robinson, M., Haywood, A., Salzmann, U., Hill, D., Sohl, L., Chandler, M., Williams, M., Foley, K., and Stoll, D.: The PRISM3D paleoenvironmental reconstruction. Stratigraphy 7, 123–139, 2010a.

Dowsett, H.J., Robinson, M.M., Foley, K.M., and Stoll, D.K.: Mid-Piacensian mean annual sea surface temperature: an analysis for data-model comparisons. Stratigraphy 7, 189–198, 2010b.

Dowsett, H.J., Foley, K.M., Stoll, D.K., Chandler, M.A., Sohl, L.E., Bentsen, M., Otto-Bliesner, B.L., Bragg, F.J., Chan, W.-L., Contoux, C., Dolan, A.M., Haywood, A.M., Jonas, J.A., Jost, A., Kamae, Y., Lohmann, G., Lunt, D.J., Nisancioglu, K.H., Abe-Ouchi, A., Ramstein, G.,

Riesselman, C.R., Robinson, M.M., Rosenbloom, N.A., Salzmann, U., Stepanek, C., Strother, S.L., Ueda, H., Yan, Q., and Zhang, Z.: Sea Surface Temperature of the mid-Piacenzian Ocean: A Data-Model Comparison. Sci. Rep. 3, 2013.

Dowsett, H., Dolan, A., Rowley, D., Moucha, R., Forte, A.M., Mitrovica, J.X., Pound, M., Salzmann, U., Robinson, M., Chandler, M., Foley, K., and Haywood, A.: The PRISM4 (mid-Piacenzian)

paleoenvironmental reconstruction. Clim. Past 12, 1519–1538, 2016.

Gent, P.R., Danabasoglu, G., Donner, L.J., Holland, M.M., Hunke, E.C., Jayne, S.R., Lawrence, D.M., Neale, R.B., Rasch, P.J., Vertenstein, M., Worley, P.H., Yang, Z.-L., and Zhang, M.: The Community Climate System Model Version 4. J. Climate 24, 4973–4991, 2011.

Guo, C., Bentsen, M., Bethke, I., Ilicak, M., Tjiputra, J., Toniazzo, T., Schwinger, J., and Otterå O.H.:

Description and evaluation of NorESM1-F: a fast version of the Norwegian Earth System Model (NorESM). Geosci. Model Dev. 12, 343–362, 2019.

Haywood, A.M., Dowsett, H.J., Otto-Bliesner, B., Chandler, M.A., Dolan, A.M., Hill, D.J., Lunt, D.J., Robinson, M.M., Rosenbloom, N., Salzmann, U., and Sohl, L.E.: Pliocene Model Intercomparison Project (PlioMIP): experimental design and boundary conditions (Experiment

1). Geosci. Model Dev. 3, 227–242, 2010.

Haywood, A.M., Hill, D.J., Dolan, A.M., Otto-Bliesner, B.L., Bragg, F., Chan, W.L., Chandler, M.A., Contoux, C., Dowsett, H.J., Jost, A., Kamae, Y., Lohmann, G., Lunt, D.J., Abe-Ouchi, A., Pickering, S.J., Ramstein, G., Rosenbloom, N.A., Salzmann, U., Sohl, L., Stepanek, C., Ueda, H., Yan, Q., and Zhang, Z.: Large-scale features of Pliocene climate: results from the Pliocene Model Intercomparison Project. Clim. Past 9, 191–209, 2013.

Haywood, A.M., Dowsett, H.J., and Dolan, A.M.: Integrating geological archives and climate models for the mid-Pliocene warm period. Nat. Commun. 7, 10646, 2016a.

Haywood, A.M., Dowsett, H.J., Dolan, A.M., Rowley, D., Abe-Ouchi, A., Otto-Bliesner, B., Chandler, M.A., Hunter, S.J., Lunt, D.J., Pound, M., and Salzmann, U.: The Pliocene Model Intercomparison Project (PlioMIP) Phase 2: scientific objectives and experimental design. Clim. Past 12, 663–675, 2016b.

Hill, D.J., Haywood, A.M., Lunt, D.J., Hunter, S.J., Bragg, F.J., Contoux, C., Stepanek, C., Sohl, L., Rosenbloom, N.A., Chan, W.L., Kamae, Y., Zhang, Z., Abe-Ouchi, A., Chandler, M.A., Jost, A., Lohmann, G., Otto-Bliesner, B.L., Ramstein, G., and Ueda, H.: Evaluating the dominant components of warming in Pliocene climate simulations. Clim. Past 10, 79–90, 2014.

Howell, F.W., Haywood, A.M., Otto-Bliesner, B.L., Bragg, F., Chan, W.L., Chandler, M.A., Contoux, C., Kamae, Y., Abe-Ouchi, A., Rosenbloom, N.A., Stepanek, C., and Zhang, Z.: Arctic sea ice simulation in the PlioMIP ensemble. Clim. Past 12, 749–767, 2016.

Hu, A., Meehl, G.A., Otto-Bliesner, B.L., Waelbroeck, C., Han, W., Loutre, M.-F., Lambeck, K., Mitrovica, J.X., and Rosenbloom, N.: Influence of Bering Strait flow and North Atlantic circulation on glacial sea-level changes. Nat. Geosci. 3, 118, 2010.

Hunter, S. J., Haywood, A. M., Dolan, A. M., and Tindall, J. C.: The HadCM3 contribution to PlioMIP Phase 2 Part 1: Core and Tier 1 experiments, Clim. Past, 15, 1691–1713, 2019

Kamae, Y., and Ueda, H.: Mid-Pliocene global climate simulation with MRI-CGCM2.3: set-up and initial results of PlioMIP Experiments 1 and 2. Geosci. Model Dev. 5, 793–808, 2012.

Kamae, Y., Yoshida, K., and Ueda, H.: Sensitivity of Pliocene climate simulations in MRI-CGCM2.3 to respective boundary conditions. Clim. Past 12, 1619–1634, 2016.

Kirkevåg, A., Iversen, T., Seland, Ø., Hoose, C., Kristjánsson, J.E., Struthers, H., Ekman, A.M.L., Ghan, S., Griesfeller, J., Nilsson, E.D., and Schulz, M.: Aerosol–climate interactions in the Norwegian Earth System Model – NorESM1-M. Geosci. Model Dev. 6, 207–244, 2013.

Li, X., Jiang, D., Zhang, Z., Zhang, R., Tian, Z., and Yan, Q.: Mid-Pliocene westerlies from PlioMIP simulations. Adv. Atmos. Sci. 32, 909–923, 2015.

Li, X., Jiang, D., Tian, Z., and Yang, Y.: Mid-Pliocene global land monsoon from PlioMIP1 simulations. Palaeogeogr. Palaeoclimatol. Palaeoecol. 512, 56–70, 2018.

Levitus, S. and Boyer, T. P.: World Ocean Atlas Volume 4: Temperature, NOAA Atlas NESDIS 4, US Government Printing Office, Washington, DC, 117, 1994.

Miller, K.G., Wright, J.D., Browning, J.V., Kulpecz, A., Kominz, M., Naish, T.R., Cramer, B.S., Rosenthal, Y., Peltier, W.R., and Sosdian, S.: High tide of the warm Pliocene: Implications of global sea level for Antarctic deglaciation. Geology 40, 407–410, 2012.

Otto-Bliesner, B.L., Jahn, A., Feng, R., Brady, E.C., Hu, A., and Löfverström, M.: Amplified North Atlantic warming in the late Pliocene by changes in Arctic gateways. Geophys. Res. Lett. 44, 957–964, 2017.

Robinson, M.M.: New quantitative evidence of extreme warmth in the Pliocene Arctic. Stratigraphy 6, 265–275, 2009.

Rosenbloom, N.: Biome4 conversion to LSM, available at: https://wiki.ucar.edu/display/paleo/Biome4+conversion+to+LSM, 2009.

Rosenbloom, N.A., Otto-Bliesner, B.L., Brady, E.C., and Lawrence, P.J.: Simulating the mid-Pliocene Warm Period with the CCSM4 model. Geosci. Model Dev. 6, 549–561, 2013.

Salzmann, U., Haywood, A.M., Lunt, D.J., Valdes, P.J., and Hill, D.J.: A new global biome reconstruction and data-model comparison for the Middle Pliocene. Global Ecol. Biogeogr. 17, 432–447, 2008.

Salzmann, U., Dolan, A.M., Haywood, A.M., Chan, W.-L., Voss, J., Hill, D.J., Abe-Ouchi, A., Otto-Bliesner, B., Bragg, F.J., Chandler, M.A., Contoux, C., Dowsett, H.J., Jost, A., Kamae, Y., Lohmann, G., Lunt, D.J., Pickering, S.J., Pound, M.J., Ramstein, G., Rosenbloom, N.A., Sohl, L., Stepanek, C., Ueda, H., and Zhang, Z.: Challenges in quantifying Pliocene terrestrial warming revealed by data–model discord. Nature Climate Change 3, 969–974, 2013.

Steele, M., Morley, R., and Ermold, W.: PHC: A global ocean hydrography with a high-quality Arctic Ocean. J. Climate, 14, 2079–2087, 2001.

Sun, Y., Ramstein, G., Contoux, C., and Zhou, T.: A comparative study of large-scale atmospheric circulation in the context of a future scenario (RCP4.5) and past warmth (mid-Pliocene). Clim.

Past 9, 1613–1627, 2013.

Tan, N., Contoux, C., Ramstein, G., Sun, Y., Dumas, C., and Sepulchre, P.: Modelling a modern-like-$pCO_2$ warm period (MIS KM5c) with two versions of IPSL AOGCM. Clim. Past Discuss. https://doi.org/10.5194/cp-2019-83, in review, 2019.

Yan, Q., Wei, T., Korty, R.L., Kossin, J.P., Zhang, Z., and Wang, H.: Enhanced intensity of global tropical cyclones during the mid-Pliocene warm period. Proc. Natl. Acad. Sci. 113, 12963–12967, 2016.

Zhang, R., Yan, Q., Zhang, Z.S., Jiang, D., Otto-Bliesner, B.L., Haywood, A.M., Hill, D.J., Dolan, A.M., Stepanek, C., Lohmann, G., Contoux, C., Bragg, F., Chan, W.L., Chandler, M.A., Jost, A.,
Kamae, Y., Abe-Ouchi, A., Ramstein, G., Rosenbloom, N.A., Sohl, L., and Ueda, H.: Mid-Pliocene East Asian monsoon climate simulated in the PlioMIP. Clim. Past 9, 2085–2099, 2013.

Zhang, Z.S., Nisancioglu, K., Bentsen, M., Tjiputra, J., Bethke, I., Yan, Q., Risebrobakken, B., Andersson, C., and Jansen, E.: Pre-industrial and mid-Pliocene simulations with NorESM-L.
Geosci. Model Dev. 5, 523−533, 2012.

Zhang, Z.S., Nisancioglu, K.H., Chandler, M.A., Haywood, A.M., Otto-Bliesner, B.L., Ramstein, G., Stepanek, C., Abe-Ouchi, A., Chan, W.L., Bragg, F.J., Contoux, C., Dolan, A.M., Hill, D.J., Jost, A., Kamae, Y., Lohmann, G., Lunt, D.J., Rosenbloom, N.A., Sohl, L.E., and Ueda, H.: Mid-pliocene Atlantic Meridional Overturning Circulation not unlike modern. Clim. Past 9,
14955–11504, 2013a.

Zhang, Z., Nisancioglu, K.H., and Ninnemann, U.S.: Increased ventilation of Antarctic deep water during the warm mid-Pliocene. Nat. Commun. 4, 1499, 2013b.

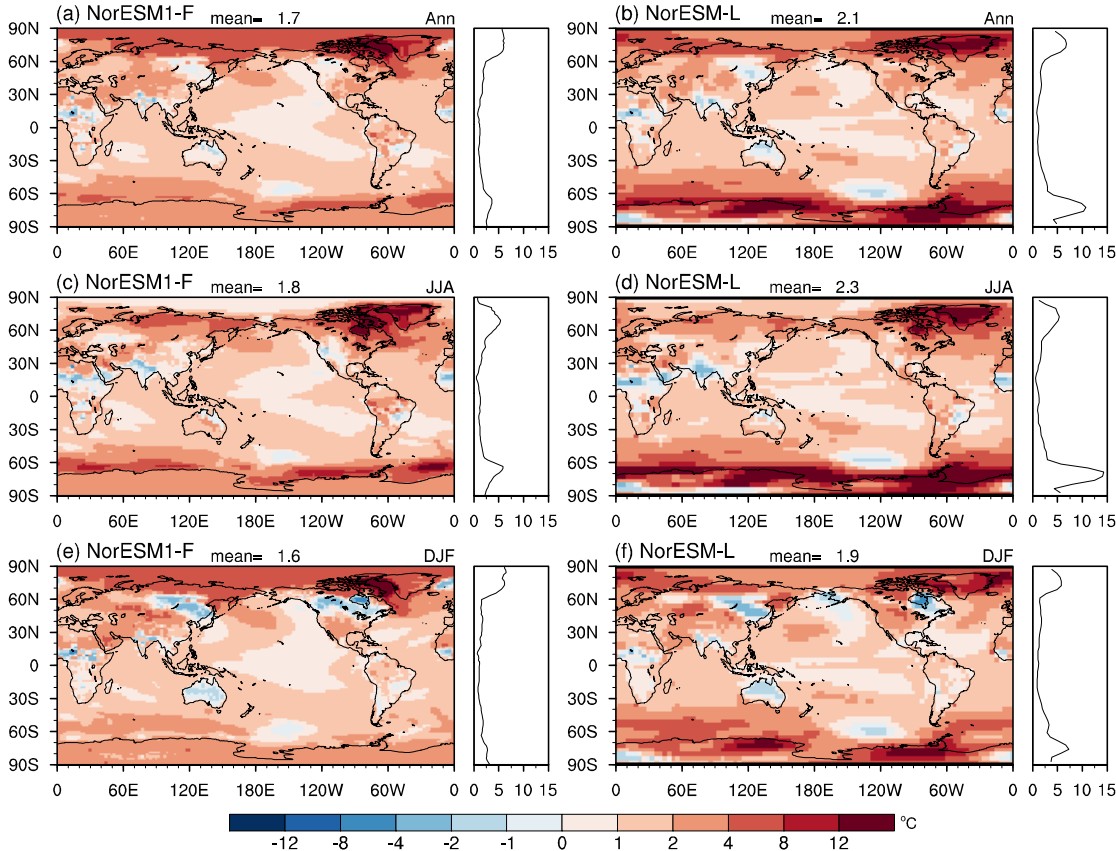

**Figure 1. The difference in climatological surface air temperatures (units: °C) between Pliocene and pre-industrial experiments according to NorESM1-F (left panel) and NorESM-L (right panel) for the annual mean (a and b), boreal summer (c and d), and boreal winter (e and f). The zonal mean is shown to the right of each plot.**

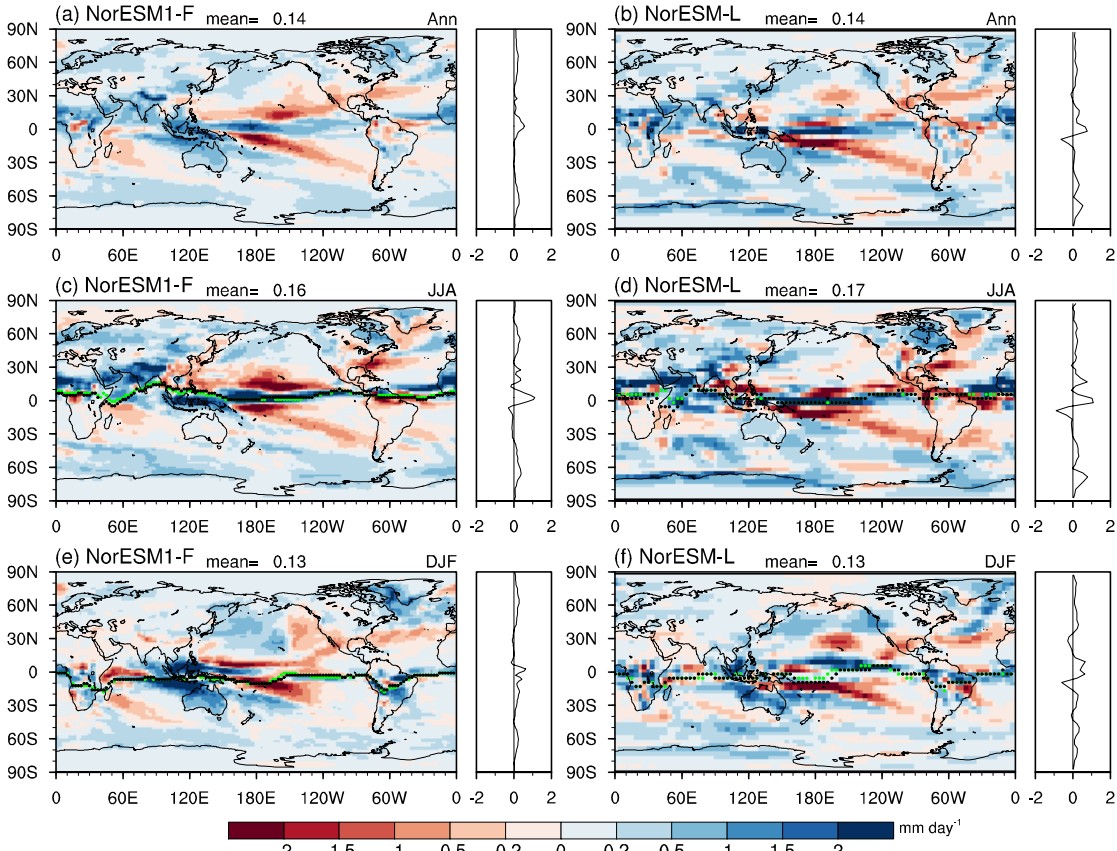

**Figure 2. Same as Figure 1 but for precipitation (units: mm day$^{-1}$). The black and green dots in c, d, e, and f indicate the positions of ITCZ in the pre-industrial and Pliocene experiments, respectively, as defined by Braconnot et al. (2007).**

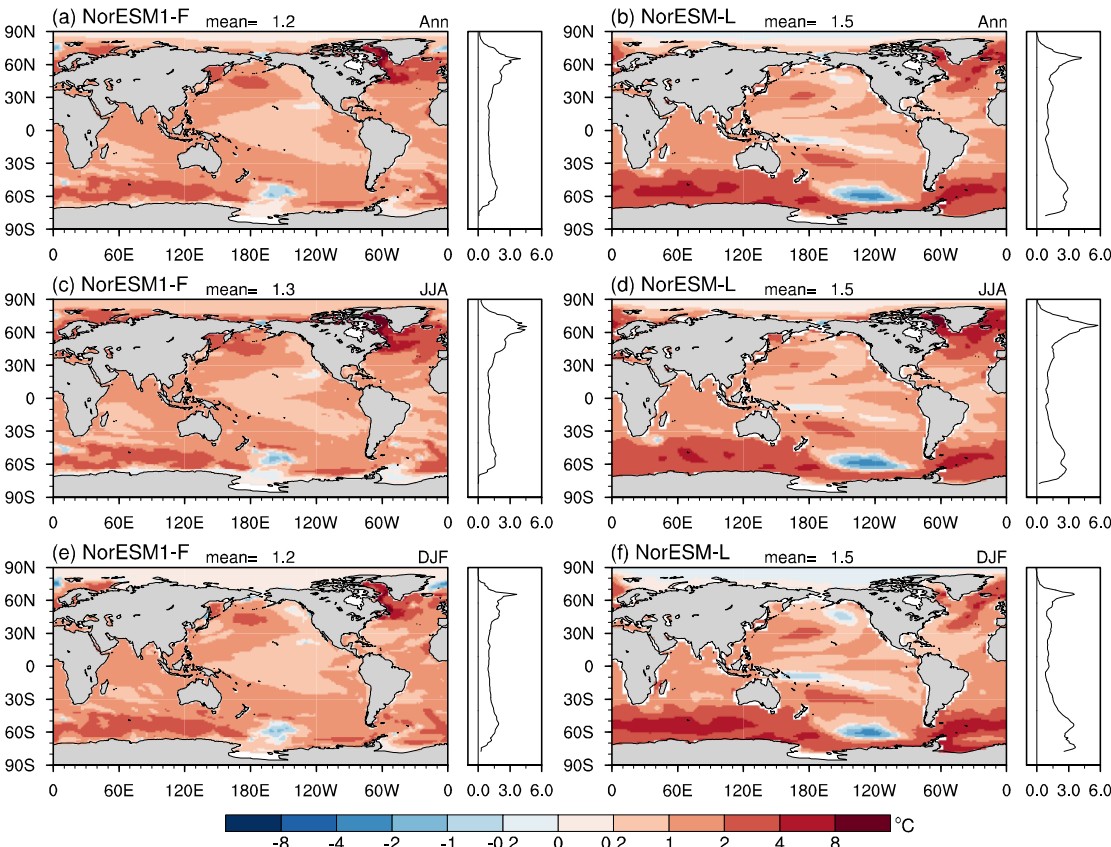

**Figure 3. Same as Figure 1 but for sea surface temperature (units: °C).**

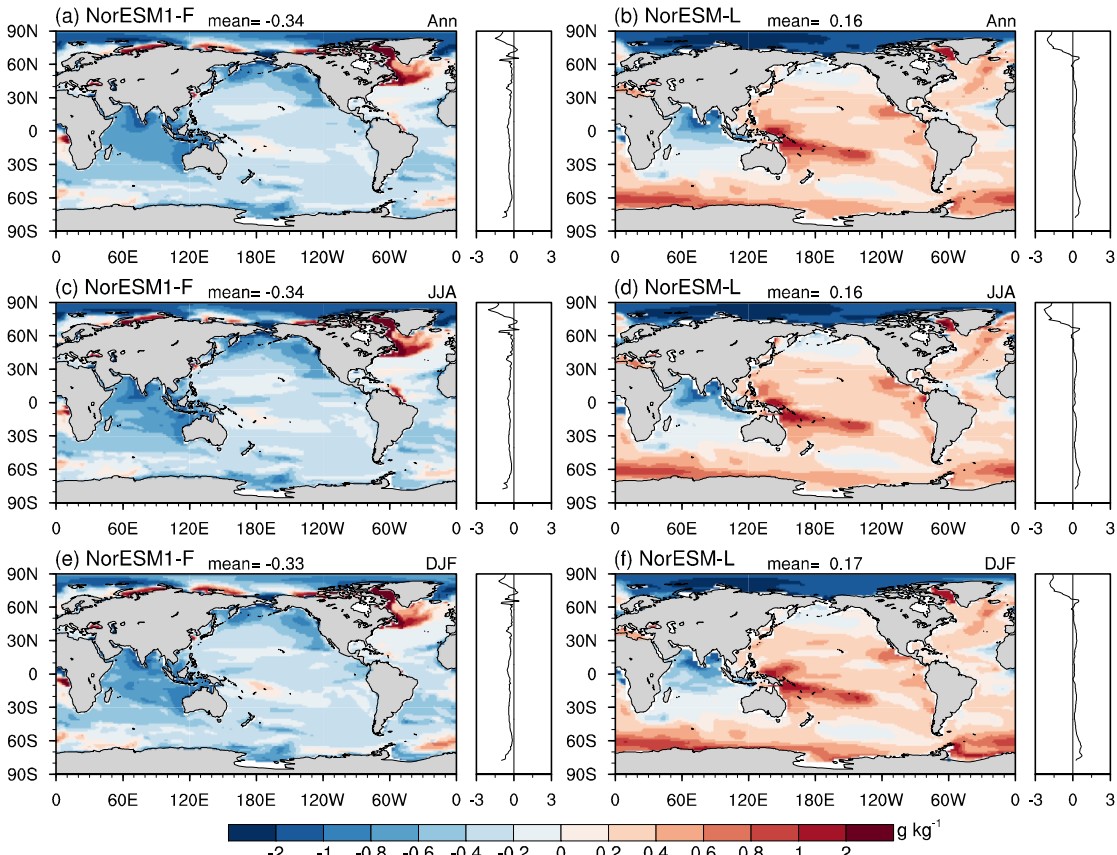

**Figure 4. Same as Figure 1 but for sea surface salinity (units: g kg⁻¹).**

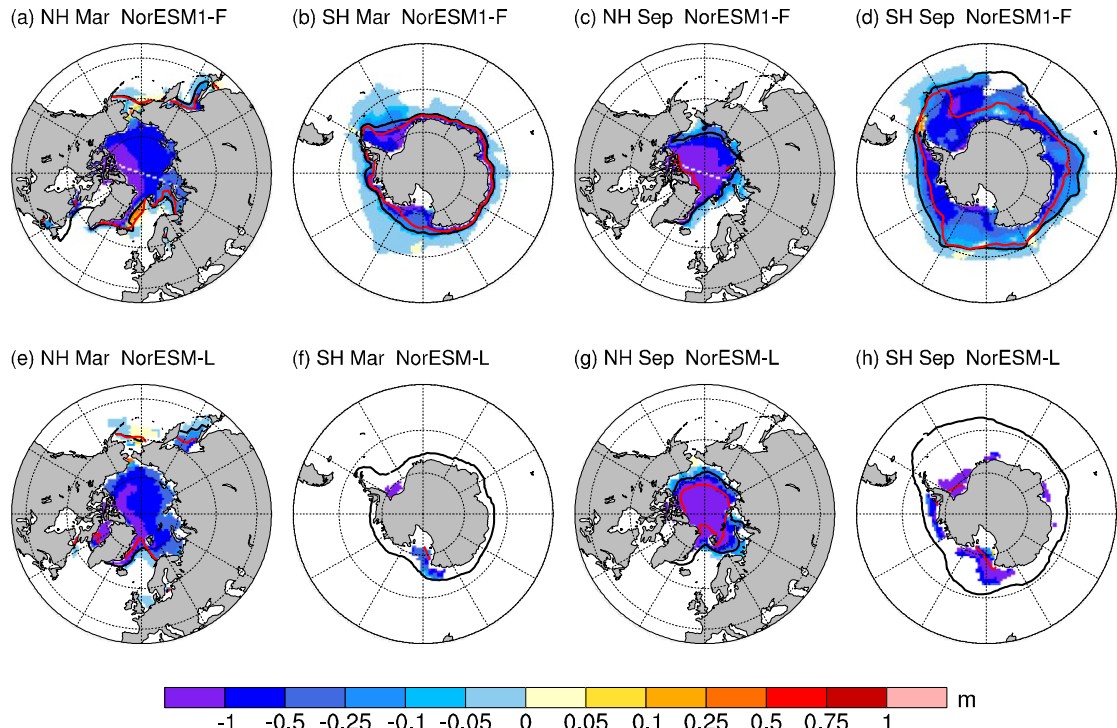

**Figure 5. The difference in climatological sea ice thickness (shading, units: m) between the Pliocene and pre-industrial experiments according to NorESM1-F (top panel) and NorESM-L (bottom panel) for March (a, b, e, and f) and September (c, d, g, and h). The red and black lines represent 15 % sea ice concentration in the Pliocene and pre-industrial experiments, respectively.**

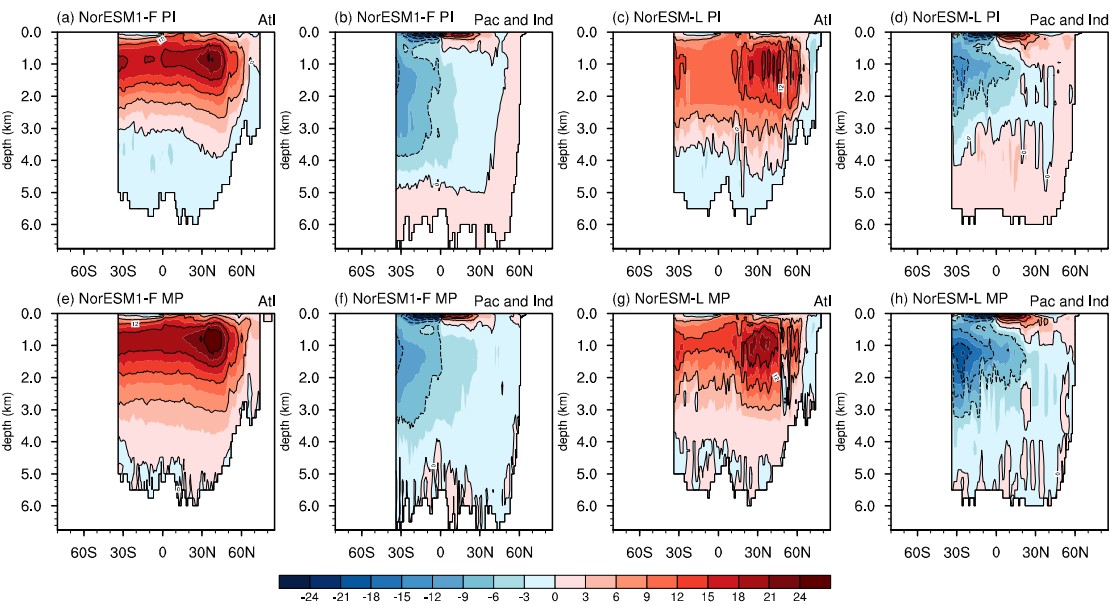

**Figure 6. Climatological meridional overturning stream functions (units: Sv) derived from the NorESM1-F (left two panels) and NorESM-L (right two panels) Pliocene and pre-industrial experiments conducted on the Atlantic Basin (a, c, e, and g) and the Pacific and Indian Ocean Basin (b, d, f, and h).**

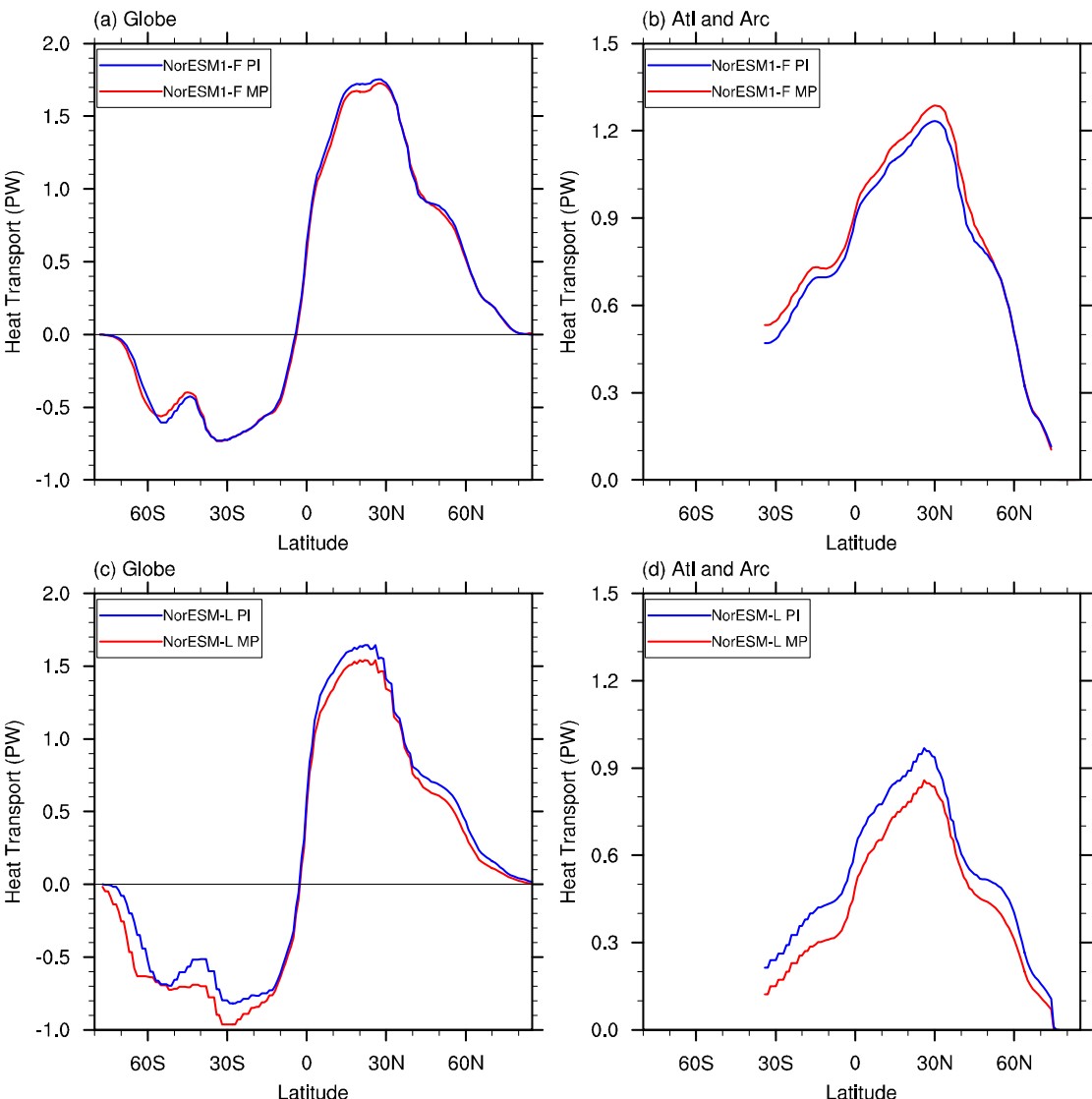

**Figure 7. Climatological meridional ocean heat transport (units: PW) derived from the Pliocene (red line) and pre-industrial (blue line) experiments according to NorESM1-F (a, b) and NorESM-L (c, d) for globe (a, c) and Atlantic and Arctic Ocean (b, d).**

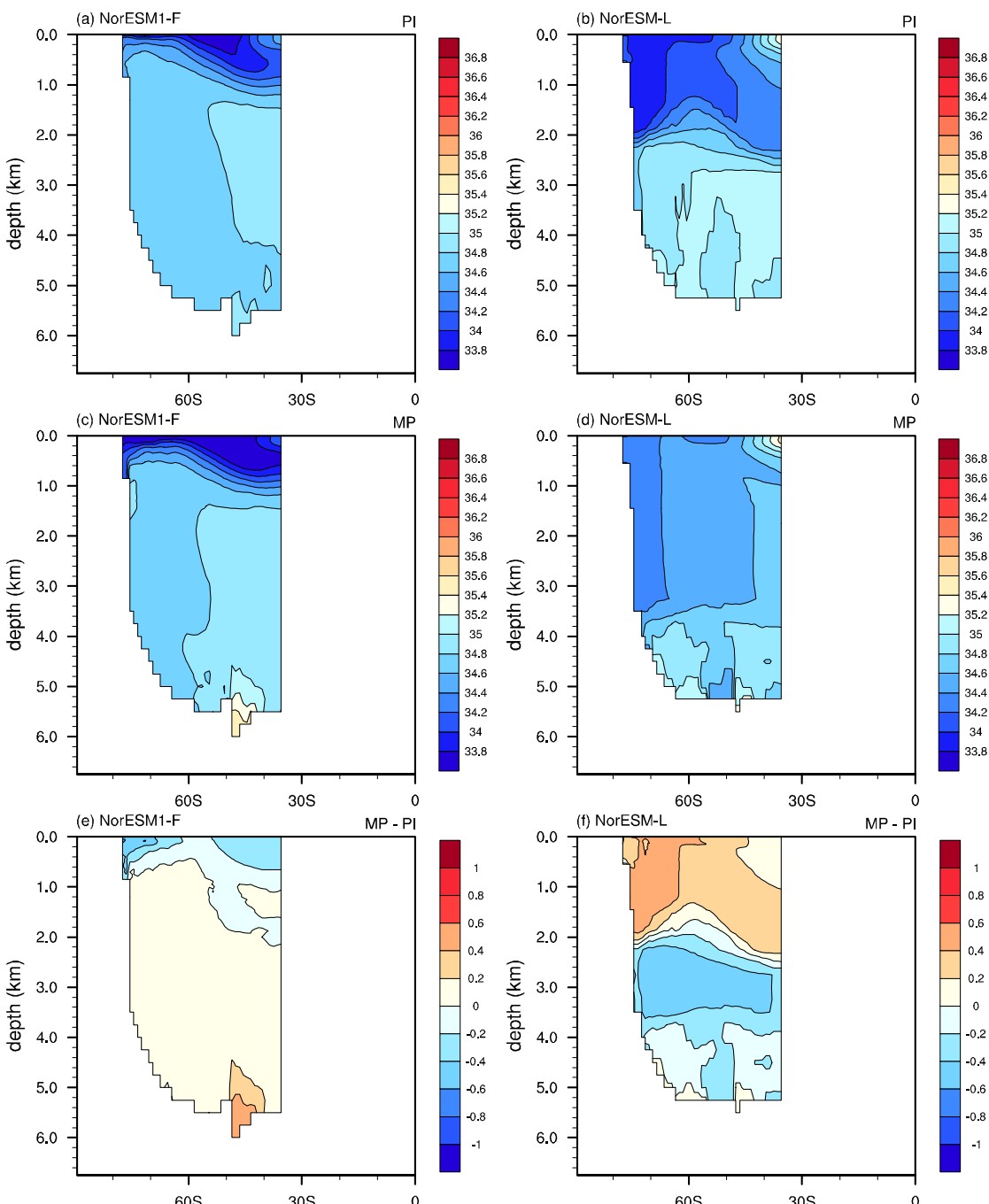

**Figure 8. Climatological Southern Ocean salinity (units: g kg⁻¹) derived from the pre-industrial (a and b) and Pliocene (c and d) experiments, and their differences (e and f) according to NorESM1-F (left panel) and NorESM-L (right panel).**

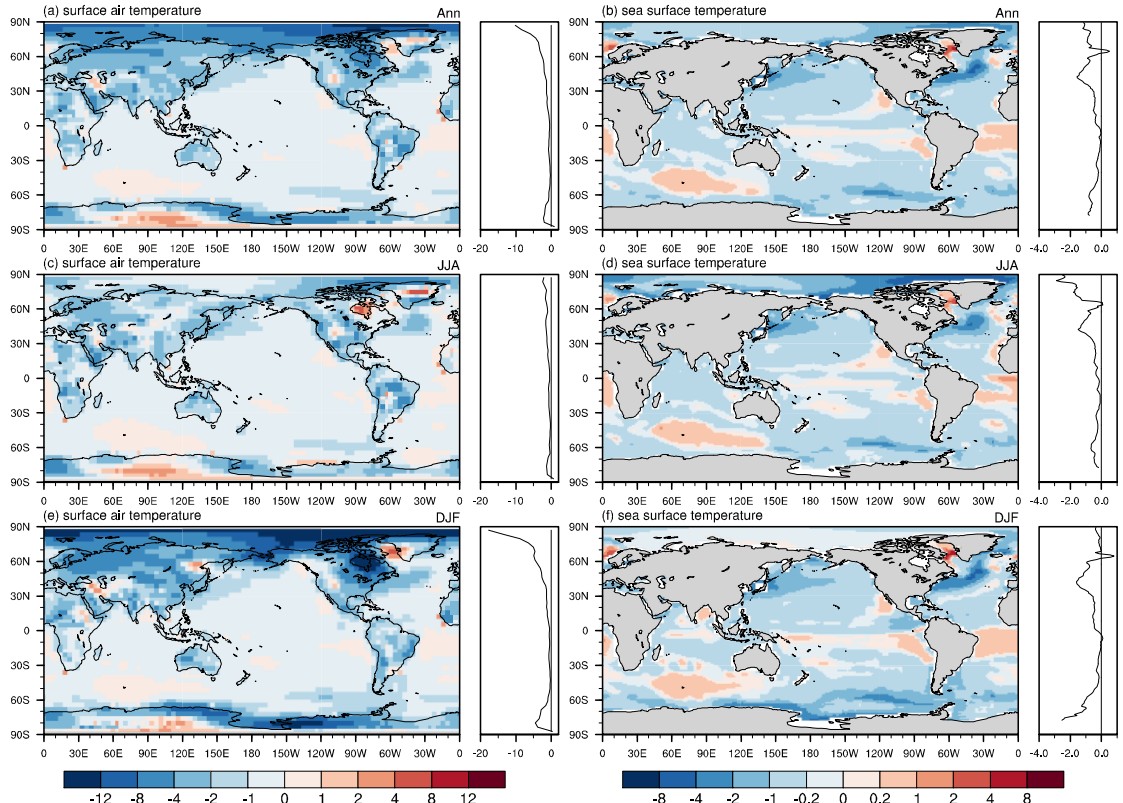

**Figure 9. Anomalies of surface air temperature change (left panel) and sea surface temperature change (right panel) (units: °C) between PlioMIP2 and PlioMIP1 (PlioMIP2 minus PlioMIP1) according to NorESM-L for the annual means (a and b), boreal summer (c and d), and boreal winter (e and f). The zonal mean is shown to the right of each plot.**

**Table 1. The model description.**

| Model version | Atmosphere | Ocean | Reference |
|---|---|---|---|
| NorESM-L | T31, ~3.75 ° | bipolar grid | Zhang et al., 2012 |
| | 26 levels in the vertical | g37, ~ 3 ° | |
| NorESM1-F | 1.9 °x2.5 ° | tripolar grid | Guo et al., 2019 |
| | 26 levels in the vertical | ~1 ° | |

**Table 2. Boundary conditions for the pre-industrial and the Pliocene experiments.**

| | Pre-industry | | Pliocene | |
|---|---|---|---|---|
| | NorESM-L | NorESM1-F | NorESM-L | NorESM1-F |
| Land-sea mask/Bathymetry | Local modern | | PRISM | |
| Topography and ice sheet height | Local modern | | Anomalies + local modern | |
| Vegetation and ice sheet cover | Local pre-industrial | | PRISM vegetation | |
| Initialization of ocean model* | Levitus T/S | PHC T/S | T/S in PlioMIP1 with NorESM-L | T/S in the previous run with 400 ppmv $CO_2$ |
| $CO_2$ (ppmv) | 280 | 284.7 | 400 | 400 |
| $N_2O$ (ppbv) | 270 | 275.68 | 270 | 275.68 |
| $CH_4$ (ppbv) | 760 | 791.6 | 760 | 791.6 |
| CFCs | 0 | 0 | 0 | 0 |
| Orbital parameters | Year 1950 | | | |
| Total integration (yr) | 2200 | 2000 | 1200 | 500** |
| Averaging period | Last 100 yr | | | |

*In the pre-industrial experiments, the ocean component of NorESM-L and NorESM1-F was both initialized from rest. The ocean component of NorESM-L was initiated from Levitus temperature and salinity (Levitus and Boyer, 1994). The ocean component of NorESM1-F was initialized from the Polar Science Center Hydrographic Climatology (PHC) 3.0, updated from Steele et al. (2001). In the Pliocene experiments, the Pliocene T/S with NorESM-L was initiated from T/S in the PlioMIP1 experiment with NorESM-L (Zhang et al., 2012). The Pliocene T/S with NorESM1-F was initiated from T/S in the pre-existed experiment under 2000 years spin-up with atmospheric $CO_2$ concentration set to 400 ppmv.

**After 2000 years spin-up with atmospheric $CO_2$ concentration set to 400 ppmv, the experiment is integrated for another 500 years forced with all Pliocene boundary conditions.

**Table 3. Regional averaged annual and seasonal anomalies between the Pliocene and pre-industrial experiments according to NorESM1-F and NorESM-L in PlioMIP2 for surface air temperature (SAT), precipitation, and sea surface temperature (SST).**

| Region | SAT (ANN / JJA / DJF, °C) | | precipitation (ANN / JJA / DJF, mm day$^{-1}$) | | SST (ANN / JJA / DJF, °C) | |
|---|---|---|---|---|---|---|
| | NorESM1-F | NorESM-L | NorESM1-F | NorESM-L | NorESM1-F | NorESM-L |
| NH high-latitude (60 °N–90 °N) | 5.2 / 4.4 / 4.9 | 4.9 / 4.5 / 3.9 | 0.18 / 0.09 / 0.21 | 0.13 /0.13 / 0.12 | 1.5 / 2.4 / 0.9 | 1.4 / 2.2 / 0.77 |
| NH middle-latitude (30 °N–60 °N) | 1.9 / 2.1 / 1.6 | 1.7 / 1.9 / 1.6 | 0.11 / 0.16 / 0.10 | 0.11 / 0.14 / 0.14 | 1.7 / 2.0 / 1.5 | 1.5 / 1.7 / 1.2 |
| NH low-latitude (0 °–30 °N) | 1.1 / 0.92 / 1.1 | 1.1 / 0.92 / 1.2 | 0.22 / 0.36 / 0.07 | 0.24 / 0.36 / 0.16 | 1.0 / 1.0 / 1.0 | 1.0 / 1.0 / 1.1 |
| SH low-latitude (30 °S–0 °) | 1.1 / 1.2 / 0.99 | 1.1 / 1.2 / 1.0 | 0.09 / -0.03 / 0.24 | 0.03 / -0.07 / 0.10 | 1.0 / 1.0 / 1.1 | 1.1 / 1.1 / 1.1 |
| SH middle-latitude (30 °S–60 °S) | 1.6 / 1.7 / 1.5 | 2.3 / 2.2 / 2.4 | 0.09 / 0.11 / 0.09 | 0.16 / 0.18 / 0.13 | 1.5 / 1.4 / 1.6 | 2.2 / 2.1 / 2.5 |
| SH high-latitude (60 °S–90 °S) | 3.2 / 4.6 / 2.1 | 7.6 / 10.3 / 4.7 | 0.20 / 0.25 / 0.14 | 0.27 / 0.38 / 0.15 | 1.0 / 0.92 / 1.2 | 2.6 / 2.5 / 2.9 |

**Table 4. Meridional overturning circulation in the Atlantic Ocean (AMOC) and the Pacific and Indian Oceans (PMOC).**

| Meridional Overturning Circulation | NorESM1-F | | | NorESM-L | | |
|---|---|---|---|---|---|---|
| | PI | MP | % | PI | MP | % |
| AMOC maximum (Sv) | 24.5 | 28.1 | 15 | 21.3 | 23.3 | 9 |
| AMOC upper cell averaged depth (m) | ~3200 | ~4700 | 47 | ~3700 | ~4800 | 30 |
| PMOC maximum above 500m, North of 5 ˚N (Sv) | 22.3 | 23.3 | 4 | 30.7 | 31.6 | 3 |
| PMOC minimum below the 500m (Sv) | –13.6 | –14.4 | 6 | –17.0 | –21.6 | 27 |

The strength of shallower circulation in the subtropical gyre of the North Pacific is represented by the PMOC maximum north of 5 ˚N above 500 m. The strength of Pacific Deep Water is represented by the PMOC minimum measured below the 500 m.

**Table 5. Similar as table 3, but for anomalies simulated according to NorESM-L between PlioMIP1 and PlioMIP2.**

| Region | SAT (ANN / JJA / DJF, ℃) | | P (ANN / JJA / DJF, mm day$^{-1}$) | | SST (ANN / JJA / DJF, ℃) | |
|---|---|---|---|---|---|---|
| | PlioMIP1 | PlioMIP2 | PlioMIP1 | PlioMIP2 | PlioMIP1 | PlioMIP2 |
| NH high-latitude (60 ℃N–90 ℃N) | 8.9 / 5.9 / 10.0 | 4.9 / 4.5 / 3.9 | 0.28 / 0.16 / 0.31 | 0.13 / 0.13 / 0.12 | 2.1 / 3.9 / 0.90 | 1.4 / 2.2 / 0.77 |
| NH middle-latitude (30 ℃N–60 ℃N) | 3.4 / 3.2 / 3.9 | 1.7 / 1.9 / 1.6 | 0.15 / 0.16 / 0.20 | 0.11 / 0.14 / 0.14 | 2.5 / 2.7 / 2.4 | 1.5 / 1.7 / 1.2 |
| NH low-latitude (0 °–30 ℃N) | 2.0 / 1.7 / 2.2 | 1.1 / 0.92 / 1.2 | 0.41 / 0.45 / 0.44 | 0.24 / 0.36 / 0.16 | 1.5 / 1.4 / 1.5 | 1.0 / 1.0 / 1.1 |
| SH low-latitude (30 ℃S–0 °) | 1.8 / 1.9 / 1.6 | 1.1 / 1.2 / 1.0 | -0.23 / -0.23 / -0.30 | 0.03 / -0.07 / 0.10 | 1.2 / 1.2 / 1.2 | 1.1 / 1.1 / 1.1 |
| SH middle-latitude (30 ℃S–60 ℃S) | 2.7 / 2.6 / 2.8 | 2.3 / 2.2 / 2.4 | 0.22 / 0.26 / 0.20 | 0.16 / 0.18 / 0.13 | 2.6 / 2.5 / 2.8 | 2.2 / 2.1 / 2.5 |
| SH high-latitude (60 ℃S–90 ℃S) | 8.9 / 11.0 / 7.1 | 7.6 / 10.3 / 4.7 | 0.29 / 0.43 / 0.12 | 0.27 / 0.38 / 0.15 | 3.3 / 3.0 / 4.1 | 2.6 / 2.5 / 2.9 |