# Peer review of "PlioMIP2 simulations with NorESM-L and NorESM1-F"

_Climate of the Past, 2019_

## Referee Comment (RC1) · Anonymous Referee #1 · 19 Oct 2019

Review of the manuscript entitled "PlioMIP2 simulations with NorESM-L and NorESM1-F" by Xiangyu Li, Chuncheng Guo, Zhongshi Zhang, Odd Helge Otterå and Ran Zhang.

This paper is a contribution to the special issue dedicated to PLIOMIP2. The authors describe the core simulations performed by 2 different versions of NORESM model.

The paper is well written and illustrated.

It fits the goal of the PLIOMIP2 special issue which is to describe the different model simulations. This manuscript is devoted to the comparison of two different versions of NorESM GCM in the common framework of PLIOMIP2 boundary conditions. In addition, for one of the model versions only, the authors provide the comparison between PLIOMIP1&2 configurations.

My general conclusion is that the study is appropriate for publication in Climate of the Past special issue after the authors answer some comments I raised below. More importantly, the authors have to clarify the inconsistency between numbers in the text and Table 3 and Fig. 1 of the paper concerning SAT. The latter mainly concern the description of what they expect from the comparison of both versions within the PLIOMIP2 framework and a more detailed discussion over northern hemisphere responses of both model versions.

- Abstract

1 The results concerning SAT and SST anomalies between mid-Pliocene and preindustrial depict a very large warming of more than 3° on the continents at global scale. Here the terrestrial warming is nearly twice as large as the ocean one. The authors should maybe emphasize on this large contrast between continent and ocean. Is this sensitivity of the warming between oceans and continents consistent with IPCCRCP simulations using NorESM versions results for ocean and land contrast?

2 The authors should if possible give the major differences between both versions to better understand the large sensitivity to the AMOC.

3 The intensification of the water cycle in a warmer climate (Clausius-Clapeyron relationship for global scale) is expected but more importantly, we would like to know whether regional patterns of precipitations are similar or not.

The abstract should briefly clarify these points..

- Section 1: Introduction

The introduction is excellent for me, summarizing the evolution of the PLIOMIP project and the contribution of the NORESM group.

- Section 2 : Model description

This section needs some improvements so that the reader may understand better the results section.

1. We need to know a bit deeper in the main text which modifications have been done in version 1-F and if they provided improvements for the preindustrial control run. Are they related with a better spatial resolution or related to the hydrologic cycle simulations?

2. What do the authors expect using the version 1-F for PLIOMIP2 with respect to what they already obtained in the previous standard (L) version? I agree that the authors referred to the paper by Guo et al (2019) for a detailed description, but we need a minimum of details for better understanding of the results described in the next section.

- Section 3: Experimental design

Why the spin up procedure is different for version 1-F?

- Section 4: Results
  - Section 4.1 Temperature

Superimposed to a zonal description, it would be interesting to discuss the result in terms of land/ocean.

The authors write: "The simulated Pliocene annual mean SAT increases by 3.2℃ (NorESM1-F) and 7.6℃ (NorESM-L) at the northern high latitudes and by 5.2℃ (NorESM1-F) and 4.9℃ (NorESM-L) at the Southern high latitudes."

This is totally inconsistent with fig. 1 and Table 3. It may be possible that the authors confused Norther and Southern hemisphere in the text. Anyway, it is crucial to clarify this point.  If you believe the text, there is a large warming over the NH for both versions, but the enhancement is much larger especially for NorESM L. It should be discussed as well as the enhanced seasonal cycle which is certainly largely responsible for the seasonal sea ice behavior in the warmer L version.

  - Section 4.3 :SST

The version 1-F depicts a smaller global warming, but a larger one at high latitude of northern hemisphere compared to the L version. It  would be interesting superimposed to the analysis which is already provided for the southern Hemisphere to add as a new section in part 5 a similar discussion concerning northern hemisphere to investigate why 1-F version is depicting a weaker response most over the globe except over mid to high latitudes of northern hemisphere. The different behavior of the two versions in north and south hemispheres should be emphasized.

  - Section 4.4 Salinity

Large differences on salinity are depicted between both versions. This result needs to be analyzed and understood at global scale and not only for the southern hemisphere.  This also points towards a new subsection in the discussion focused on Northern hemisphere.

  - Section 4.5:  Sea ice

 An important result concerns the summer arctic sea ice especially with regard to future climate. It could be interesting, in the discussion, to add more details on the causes of these differences.

- Section 5: Discussion

This part includes 2 sections. The first one could be improved and enlarged because it deals mainly with the Southern Hemisphere.

In the discussion section, the part concerning south hemisphere salinity difference and its relation with ocean dynamics and sea ice is appropriate but we expect a bit more discussion on topics I raised above, especially on differences between both versions on NH high latitudes for land and sea thermal contrast  and arctic sea ice and AMOC/PMOC responses.

The second part of the discussion raises an important point on sensitivity of different models to the closure of seaways and the authors should point out also in their conclusions that these new results, because they are different from the previous synthesis provided by Shang for PLIOMIP1, could certainly be an important focus of the future intercomparison within PLIOMIP2 project.

Figures and Tables: The numbers in Table 3 are compatible with Fig. 1, but the text is not constituent with them. Concerning Table 3 and section 4.3 (SST), it is difficult to know from the table if the numbers in the text are correct.

---

## Editor Comment (EC1) · Wing-Le Chan (Editor) · 23 Oct 2019

**Comments from the editor to the authors of the manuscript entitled "PlioMIP2 simulations with NorESM-L and NorESM1-F"**

It would be interesting to know what exactly in the two models is causing them to show such contrasting salinities. The difference in the signs of the sea surface salinity change really stood out.

Below, I list a (non-exhaustive) list of non-technical corrections the authors should consider:

Page 1, line 23: become → becomes

Page 2, line 7: The global mean sea level was higher than that of today, with a peak of

$22\pm10$ meters (Miller et al., 2012)

Page 5, line 20: … similar pattern to those…

Page 5, line 21: …in Circum-Arctic regions…

Page 5, line 27: no comma after 'of'

Page 6, line 18: …except for Baffin Bay…

Page 7, line 21: -21.6 for the Pliocene experiment…

Page 7, line 28: …versions is likely to be associated…

Page 8, line 3:   Add 'Therefore' to be beginning of the sentence so that it links with the previous statement.

Page 8, line 6: The presence of less sea ice leads to a reduction in albedo and to a more active ocean-atmosphere interaction, contributing to…

Page 8, line 8: …ventilation in the Southern Ocean…

Page 8, line 22: appears  →  appear

Page 10, line 4: …testing the impacts of the differences between the boundary conditions of PlioMIP1 and those of PlioMIP2, ie…

Page 10, line 10: requests  →  request

Tables 3 and 5:   The numbers are written in groups of 3 (ANN, JJA and DJF). It may be easier to visualise and to compare if the authors inserted a gap on either side of the slashes, or put each number on a separate line.

---

## Referee Comment (RC2) · Anonymous Referee #2 · 2 Nov 2019

Review of "PlioMIP2 simulations with NorESM-L and NorESM1-F" by Xiangyu Li, Chuncheng Guo, Zhongshi Zhang, Odd Helge Ottera, and Ran Zhang

This manuscript presents initial results from the Nor-ESM modeling group for two simulations of the mid-Pliocene Warm Period experiment (Eoi400 of Haywood et al., 2016b), the core paleoclimate experiment of the Pliocene Model Intercomparison Project (PlioMIP) Phase 2, as a contribution to CMIP6. The simulations were run with the older NorESM-L (also used in PlioMIP1) and the more recent NorESM1-F. Six key diagnostic variables were examined, and the NorESM-L PlioMIP2 run was also compared to the group's earlier efforts for PlioMIP Phase 1. This is a solid contribution to the PlioMIP2 effort, and I recommend publication of this paper, subject to minor modifications to address the comments raised here.

General comments:

[Figure]

Of note is the authors' finding that the NorESM1-F mid-Pliocene simulation actually warms less (+1.7 degC global mean SAT, +1.2 degC global mean annual SST compared to PI control) than the equivalent simulation with the older NorESM-L model (+2.1degC global mean SAT, +1.5 degC global mean annual SST). This relative cooling of higher resolution model compared to the lower resolution model is not entirely expected, nor is both model versions' relative cooling of the Pliocene simulation compared to PI control; it is also not consistent with some of the other PlioMIP2 experiments already reported (MRI-CGCM2.3, CCSM4, IPSL-CM5A). Furthermore, the NorESM1-F Eoi400 simulation is itself cooler than the equivalent PlioMIP1 simulation (-1.1 degC global mean SAT). The authors attribute this primarily to the change in paleogeographic boundary conditions from PlioMIP1 to PlioMIP2. While this is certainly possible for NorESM-L, paleogeography alone cannot address why the newer NorESM1-F is generally not as warm as the older NorESM-L under PlioMIP2 boundary conditions.

It would be helpful to know how the equilibrium climate sensitivity differs between the two model versions. It would also be useful to know whether the authors have previously documented any differences in run results owing the horizontal grid resolution differences between the coarser grid NorESM-L and finer grid NorESM1-F.

Specific comments:

Page 3, lines 17-19 – Can the authors be more specific about the additional improvements to the MICOM ocean component of NorESM? Also, the authors note that NorESM1-F was run without the CAM4-Oslo advanced scheme for interactions between aerosols and clouds. Is there a CMIP6 PI control run available for NorESM1-F with the CAM4-Oslo scheme enabled, to compare with the NorESM1-F described here? I wonder whether the absence of this scheme with the newer model might also contribute to the lower magnitude of warming in the mid-Pliocene run described here.

Page 4, lines 10-15 – It is unclear why NorESM-L was run with different PI greenhouse gas values (280 ppmv, 270 ppbv, 760 ppbv of CO2, N2O and CH4 respectively) compared to NorESM1-F (284.7 ppmv, 275.68 ppbv, 791.6 ppbv of CO2, N2O and CH4 respectively) for the PI control run. If this was done so that the only key difference between PlioMIP1 and PlioMIP2 simulations with NorESM-L was the new paleogeographic reconstruction for PlioMIP2, it would be helpful to clarify that. If, however, the code for NorESM-L has been updated since 2012, it would be necessary to provide more detail on what has changed.

Section 4.5 Sea Ice, pages 6-7 – The sea ice differences between NorESM-L and NorESM1-F merit some additional discussion, especially for the Southern Ocean around Antarctica. Can the authors elaborate on why NorESM1-F is producing so much more ice in this region?

Technical comments:

Page 2, line 21 – There is a reference here to Zhang, R. et al 2013. Elsewhere, there are cites for Zhang, R. et al 2013a and Zhang, R. et al 2013b, but there are a total of three Zhang, R. et al 2013 references in the reference list. These should be renumbered to avoid confusion.

Page 4, line 4 – How many ocean layers for NorESM1-F?

Page 5, lines 15-17 – The description of the regional temperature highs is not consistent with Table 3 – perhaps because Table 3 lists regions from SH pole to NH pole, which is a little non-intuitive.

Page 5, line 21 – Should read "circum-Arctic" rather than "circus-Arctic"

Page 9, lines 6-7 – Perhaps say "In contrast," rather than "On the contrary,"

Table 3 and Table 5 – listing regions starting with the NH polar region at the top would be a more intuitive way to present this information

---

## Author Comment (AC1) · 29 Nov 2019

*The comments are in blue, and our responses are in black.*

Response to Editor:

Comments from the editor

It would be interesting to know what exactly in the two models is causing them to show such contrasting salinities. The difference in the signs of the sea surface salinity change really stood out.

In our experimental flow (Figure Sketch), there are divergent responses in global mean sea surface salinity (SSS) in PlioMIP2 experiment with NorESM1-F and NorESM-L. There is a slight positive shift in global mean SSS in the NorESM-L simulation, and a negative shift in global mean SSS in the NorESM1-F simulation (Note the mean value in Fig. 4). The divergent responses are likely associated with the different vertical redistribution of salt in the two models, due to differences in e.g. surface layer mixing, ocean ventilation, convection and circulation. The two models have different vertical resolutions and horizontal/vertical mixing schemes, which makes it difficult to disentangle the factors causing the contrasting salinity responses.

[Figure]

**NorESM1-F**

PI control experiment / 2000 yrs

(PHC T/S initialization)

400 ppmv $CO_2$ experiment / 2000 yrs → PlioMIP2 experiment / 500 yrs

**NorESM-L**

PI control / 2200 yrs

(Levitus T/S initialization)

PlioMIP1 experiment / 1500 yrs → PlioMIP2 experiment / 1200 yrs

**Figure Sketch for NorESM1-F and NorESM-L experiments flow.**

However, when the shift in global mean SSS is removed, NorESM-L and NorESM1-F show similar regional anomalies. Both versions show that the SSS contrast among the Indian Ocean, the Arctic and the rest of the oceans is intensified in the Pliocene experiment (Fig. S2). We added those in Section 4.4 in the manuscript with the revisions marked.

[Figure]

**Fig. 4.** The difference in climatological sea surface salinity (units: g kg$^{-1}$) between Pliocene and pre-industrial experiments according to NorESM1-F (left panel) and NorESM-L (right panel) for the annual mean (a and b), boreal summer (c and d), and boreal winter (e and f). The zonal mean is shown to the right of each plot.

[Figure]

**Fig. S2. Same as Figure 1, but for each grid, the global mean shift is excluded to emphasize the response of the sea surface salinity contrast between ocean basins in the Pliocene experiment.**

Below, I list a (non-exhaustive) list of non-technical corrections the authors should consider:

Page 1, line 23: become → becomes

Page 2, line 7: The global mean sea level was higher than that of today, with a peak of 22±10 meters (Miller et al., 2012)

Page 5, line 20: … similar pattern to those…

Page 5, line 21: …in Circum-Arctic regions…

Page 5, line 27: no comma after 'of'

Page 6, line 18: …except for Baffin Bay…

Page 7, line 21: -21.6 for the Pliocene experiment…

Page 7, line 28: …versions is likely to be associated…

Page 8, line 3: Add 'Therefore' to be beginning of the sentence so that it links with the previous statement.

Page 8, line 6: The presence of less sea ice leads to a reduction in albedo and to a more active ocean-atmosphere interaction, contributing to…

Page 8, line 8: …ventilation in the Southern Ocean…

Page 8, line 22: appears →appear

Page 10, line 4: …testing the impacts of the differences between the boundary

conditions of PlioMIP1 and those of PlioMIP2, ie…

Page 10, line 10: requests →request

Tables 3 and 5: The numbers are written in groups of 3 (ANN, JJA and DJF). It may be easier to visualise and to compare if the authors inserted a gap on either side of the slashes, or put each number on a separate line.

Revised. Thanks for the improvement to our manuscript.

---

## Author Comment (AC2) · 29 Nov 2019

Point to point response to reviewers' comments

*The comments are in blue, and our responses are in black.*

Response to Reviewer 1:

Comments from Reviewer 1

This paper is a contribution to the special issue dedicated to PLIOMIP2. The authors describe the core simulations performed by 2 different versions of NORESM model.

The paper is well written and illustrated.

It fits the goal of the PLIOMIP2 special issue which is to describe the different model simulations. This manuscript is devoted to the comparison of two different versions of NorESM GCM in the common framework of PLIOMIP2 boundary conditions. In addition, for one of the model versions only, the authors provide the comparison between PLIOMIP1&2 configurations.

My general conclusion is that the study is appropriate for publication in Climate of the Past special issue after the authors answer some comments I raised below. More importantly, the authors have to clarify the inconsistency between numbers in the text and Table 3 and Fig. 1 of the paper concerning SAT. The latter mainly concern the description of what they expect from the comparison of both versions within the PLIOMIP2 framework and a more detailed discussion over northern hemisphere responses of both model versions.

We apologize for the confusion of Northern and Southern hemispheres in the text. The numbers in tables 3 and 5 are right. We have corrected the related numbers in the text in the revised version. Further, we modified the description of the statistics in table 3 and 5 with order from the Northern pole to the Southern pole.

We thank the reviewer for the thorough assessment and constructive comments on our manuscript. We respond to the reviewer's comments below.

Specific comments
- Abstract
1 The results concerning SAT and SST anomalies between mid-Pliocene and preindustrial depict a very large warming of more than 3 ° on the continents at global scale. Here the terrestrial warming is nearly twice as large as the ocean one. The authors should maybe emphasize on this large contrast between continent and ocean. Is this sensitivity of the warming between oceans and continents consistent with IPCCRCP simulations using NorESM versions results for ocean and land contrast?

Revised. We emphasized the warming contrast between continent and ocean in

both abstract and result. In the revised manuscript, we added text of "with a greater warming over land than over ocean" in the abstract (Page 1, Line 23–24 in the manuscript with the revisions marked).

We also added sentences in Section 4.1: "Both NorESM1-F and NorESM-L simulate stronger warming over land than over ocean. Relative to the pre-industrial period, the simulated Pliocene global mean surface air temperature (SAT) over land increases by 2.3 ℃ with NorESM-L and 2.0 ℃ with NorESM1-F, which is notably larger than the warming over ocean (2.0 ℃ and 1.6 ℃ for the NorESM-L and the NorESM1-F, respectively). This stronger warming over land is a common feature in most the PlioMIP2 simulations. However, the simulated zonal mean SAT over land is nearly twice as large as in the ocean at the northern high latitudes (Fig. S1).".

[Figure]

**Fig. S1. The zonal mean of the difference in climatological annual mean surface air temperatures (units: ℃) between Pliocene and pre-industrial experiments according to NorESM1-F (left panel) and NorESM-L (right panel). The black, red, and blue lines represent values over globe, land, and ocean, respectively.**

The warming sensitivity between the oceans and the continents in the Pliocene simulation is consistent with that found in RCP2.6 and RCP8.5 simulations with NorESM1-M (see Figure 8 in Iversen et al., 2013).

2 The authors should if possible give the major differences between both versions to better understand the large sensitivity to the AMOC.

Revised.

We added the following sentences in the abstract: "NorESM1-M is the version of NorESM that contributed to the Coupled Model Intercomparison Project Phase 5 (CMIP5). NorESM-L is the low-resolution of NorESM1-M, whereas NorESM1-F is a computationally efficient version of NorESM1-M, with similar resolutions and updated physics. Relative to NorESM1-M, there are notable improvements in simulating the strength of the AMOC and the distribution of sea ice in NorESM1-F,

partly due to the updated ocean physics." (Page 1, Line 18–22 in the manuscript with the revisions marked.)

3 The intensification of the water cycle in a warmer climate (Clausius-Clapeyron relationship for global scale) is expected but more importantly, we would like to know whether regional patterns of precipitations are similar or not.

Revised.

We added and re-organized some sentences in the abstract: "The simulated precipitation for the Pliocene increases by 0.14 mm day$^{-1}$ globally in both model versions, with large increases in the tropics and especially in the monsoon regions and only minor changes, or even slight decreases, in subtropical regions. The intertropical convergence zone (ITCZ) shifts northward in the Atlantic and Africa in boreal summer." (Page 1, Line 26–28 and Page 2, Line 1–2 in the manuscript with the revisions marked.)

The abstract should briefly clarify these points.

Revised. Thanks for these suggestions.

● Section 1: Introduction

The introduction is excellent for me, summarizing the evolution of the PLIOMIP project and the contribution of the NORESM group.

We thank the reviewer for the positive remarks.

● Section 2: Model description

This section needs some improvements so that the reader may understand better the results section.

1. We need to know a bit deeper in the main text which modifications have been done in version 1-F and if they provided improvements for the preindustrial control run. Are they related with a better spatial resolution or related to the hydrologic cycle simulations?

2. What do the authors expect using the version 1-F for PLIOMIP2 with respect to what they already obtained in the previous standard (L) version? I agree that the authors referred to the paper by Guo et al (2019) for a detailed description, but we need a minimum of details for better understanding of the results described in the next section.

We add some more details about NorESM1-F in section 2 in the revised manuscript.

Compared to NorESM1-M, NorESM1-F takes some measures to improve computational performance, employs several physical updates and parameterization modifications in ocean and atmosphere components.

Paleoclimate simulations with coupled model, e.g., NorESM-L, often require thousands of years' integration to reach equilibrium, usually at the expense of

resolution to save computational resources. Compared to NorESM-L, NorESM1-F has several advantages in paleoclimate modelling, such as the higher resolution, faster running speed, and several improvements, such as more realistic AMOC, sea ice distribution and hydrological cycle.

Section 3: Experimental design

Why the spin up procedure is different for version 1-F?

Before we ran the Pliocene experiment with NorESM1-F, there was one previous simulation with atmosphere $CO_2$ set at 400 ppmv spin-up for 2000 years using NorESM1-F (Figure Sketch). In this spin-up experiment, the topography was not changed to the PlioMIP2 conditions. Initialized from this experiment, the Pliocene experiment with NorESM1-F is integrated for another 500 years long. 500-year is the minimum integration length for PlioMIP2 simulation (Haywood et al., 2016).

We added the details about the initialization of ocean model in Table 2 in the revised manuscript.

[Figure]

**Figure Sketch for NorESM1-F and NorESM-L experiments flow.**

Section 4: Results

○ Section 4.1 Temperature

Superimposed to a zonal description, it would be interesting to discuss the result in terms of land/ocean.

Revised. Please see our previous response to the reviewer's specific comment regarding the warming contrast between continent and ocean.

The authors write: "The simulated Pliocene annual mean SAT increases by 3.2°C (NorESM1-F) and 7.6°C (NorESM-L) at the northern high latitudes and by 5.2°C (NorESM1-F) and 4.9°C (NorESM-L) at the Southern high latitudes."

This is totally inconsistent with fig. 1 and Table 3. It may be possible that the

authors confused Norther and Southern hemisphere in the text. Anyway, it is crucial to clarify this point. If you believe the text, there is a large warming over the NH for both versions, but the enhancement is much larger especially for NorESM L. It should be discussed as well as the enhanced seasonal cycle which is certainly largely responsible for the seasonal sea ice behavior in the warmer L version.

We apologize for the confusion of Northern and Southern hemispheres in the text. The numbers in tables 3 and 5 are right. We have corrected the related numbers in the text in the revised version. Further, we modified the description of the statistics in table 3 and 5 with order from the Northern pole to the Southern pole, following the other reviewer's suggestion.

We add the discussion on the seasonal warming and sea ice reduction in section 5.1. We have re-organized the related sentences as: "On the one hand, the larger seasonal warming in the Southern Ocean favors less sea ice extent in the Pliocene experiment simulated with NorESM-L. On the other hand, the presence of less sea ice, leads to a reduction in albedo and to a more active ocean-atmosphere interaction, and contributes to the higher levels of Southern Ocean warming in the Pliocene experiment simulated with NorESM-L." (Page 9, Line 13–17 in the manuscript with the revisions marked).

○ Section 4.3 :SST

The version 1-F depicts a smaller global warming, but a larger one at high latitude of northern hemisphere compared to the L version. It would be interesting superimposed to the analysis which is already provided for the southern Hemisphere to add as a new section in part 5 a similar discussion concerning northern hemisphere to investigate why 1-F version is depicting a weaker response most over the globe except over mid to high latitudes of northern hemisphere. The different behavior of the two versions in north and south hemispheres should be emphasized.

We added one paragraph to discuss the warming dissimilarities of the two NorESM versions at the northern middle and high latitudes in section 5.1. Please see Section 5.1 in revised manuscript.

○ Section 4.4 Salinity

Large differences on salinity are depicted between both versions. This result needs to be analyzed and understood at global scale and not only for the southern hemisphere. This also points towards a new subsection in the discussion focused on Northern hemisphere.

In our experimental flow (Figure Sketch), there are divergent responses in global mean sea surface salinity (SSS) in PlioMIP2 experiment with NorESM1-F and NorESM-L. There is a slight positive shift in global mean SSS in the NorESM-L simulation, and a negative shift in global mean SSS in the NorESM1-F simulation (Note the mean value in Fig. 4). The divergent responses are likely associated with the

different vertical redistribution of salt in the two models, due to differences in e.g. surface layer mixing, ocean ventilation, convection and circulation. The two models have different vertical resolutions and horizontal/vertical mixing schemes, which makes it difficult to disentangle the factors causing the contrasting salinity responses.

However, when the shift in global mean SSS is removed, NorESM-L and NorESM1-F show similar regional anomalies. Both versions show that the Both versions show that the SSS contrast among the Indian Ocean, the Arctic and the rest of the oceans is intensified in the Pliocene experiment (Fig. S2). We added those in Section 4.4 in the manuscript with the revisions marked.

[Figure]

**Fig. 4. The difference in climatological sea surface salinity (units: g kg$^{-1}$) between Pliocene and pre-industrial experiments according to NorESM1-F (left panel) and NorESM-L (right panel) for the annual mean (a and b), boreal summer (c and d), and boreal winter (e and f). The zonal mean is shown to the right of each plot.**

[Figure]

**Fig. S2. Same as Figure 1, but for each grid, the global mean shift is excluded to emphasize the response of the sea surface salinity contrast between ocean basins in the Pliocene experiment.**

Apparently, the sea surface salinity increase in the Atlantic is larger with NorESM1-F than in NorESM-L. We added sentences in Section 5.1: "In associated with the larger salinity increase in the northern North Atlantic (Fig. 4), the enhancement of AMOC is larger with NorESM1-F than with NorESM-L (~15% vs. ~9%), which favors the larger responses in the Pliocene northward ocean heat transport to the Atlantic with NorESM1-F (Figs. 6 and 7)." (Page 10, Line 12–15 in manuscript with the revisions marked).

○ Section 4.5: Sea ice

An important result concerns the summer arctic sea ice especially with regard to future climate. It could be interesting, in the discussion, to add more details on the causes of these differences.

We added the discussion about sea ice in the Section 5.1.

For instance we re-organized some sentences in Section 5.1: "Simulated Pliocene southward ocean heat transport to the Southern Ocean is reduced according to NorESM1-F, but increased according to NorESM-L (Fig. 7), which partly explains the reduction in the Southern Ocean sea ice extent being more pronounced for NorESM-L than it is according to NorESM1-F (Fig. 5) (Page 9, Line 12–15 in the manuscript with the revisions marked)."

We also added the following sentences in Section 5.1: "The stronger Pliocene warming at the northern high latitudes is most likely related to the mechanism responsible for the larger responses in sea ice reduction with NorESM1-F, since the clear sky albedo, particularly in sea ice regions, dominates the high latitudes warming in Pliocene (Hill et al., 2014). In associated with the larger salinity increase in the northern North Atlantic (Fig. 4), the enhancement of AMOC is larger with NorESM1-F than with NorESM-L (~15% vs. ~9%), which favors the larger increase in the Pliocene northward ocean heat transport to the Atlantic with NorESM1-F (Figs. 6 and 7). Correspondingly, the less sea ice simulated in the Pliocene experiment contributes to a larger warming at the high latitudes with NorESM1-F than with NorESM-L through the ice-albedo feedback (Figs. 1 and 3)." (Page 10, Line 5–13 in the manuscript with the revisions marked).

- Section 5: Discussion

This part includes 2 sections. The first one could be improved and enlarged because it deals mainly with the Southern Hemisphere.

In the discussion section, the part concerning south hemisphere salinity difference and its relation with ocean dynamics and sea ice is appropriate but we expect a bit more discussion on topics I raised above, especially on differences between both versions on NH high latitudes for land and sea thermal contrast and arctic sea ice and AMOC/PMOC responses.

In Section 5.1, we added a discussion about the difference in the simulated Pliocene warming at the northern middle and high latitudes. Here, we discussed the ocean heat transport, sea ice, AMOC, and salinity change. Please see Section 5.1 in the manuscript with the revisions marked.

The land sea thermal contrast is not discussed here since both versions simulate similar land/sea thermal contrasts in terms of zonal means (Fig. S1). We already added a related description of the land sea thermal contrast in Section 4.1. Please see Page 6, Line 14–20 in the manuscript with the revisions marked.

The second part of the discussion raises an important point on sensitivity of different models to the closure of seaways and the authors should point out also in their conclusions that these new results, because they are different from the previous synthesis provided by Shang for PLIOMIP1, could certainly be an important focus of the future intercomparison within PLIOMIP2 project.

Revised. In the conclusions, we added the following sentence: "The model–dependent sensitivity to the closure of the ocean gateways in the northern high latitudes will be an interesting question that is worth further attention within the PlioMIP2 community.". (Page 12, Line 5–7 in the manuscript with the revisions marked).

Figures and Tables: The numbers in Table 3 are compatible with Fig. 1, but the

text is not constituent with them. Concerning Table 3 and section 4.3 (SST), it is difficult to know from the table if the numbers in the text are correct.

We confused the Northern and Southern hemispheres in the text. The numbers in tables 3 and 5 are right. In the revised version, we modified the description of the statistics in table 3 and 5 with order from the Northern pole to the Southern pole.

Revised. Please see Tables 3 and 5.

Referrences

Guo, C., Bentsen, M., Bethke, I., Ilicak, M., Tjiputra, J., Toniazzo, T., Schwinger, J., and Otterå, O.H.: Description and evaluation of NorESM1-F: a fast version of the Norwegian Earth System Model (NorESM). Geosci. Model Dev. 12, 343–362, 2019.

Haywood, A.M., Dowsett, H.J., Dolan, A.M., Rowley, D., Abe-Ouchi, A., Otto-Bliesner, B., Chandler, M.A., Hunter, S.J., Lunt, D.J., Pound, M., and Salzmann, U.: The Pliocene Model Intercomparison Project (PlioMIP) Phase 2: scientific objectives and experimental design. Clim. Past 12, 663–675, 2016.

Iversen, T., Bentsen, M., Bethke, I., Debernard, J. B., Kirkevåg, A., Seland, Ø., Drange, H., Kristjansson, J. E., Medhaug, I., Sand, M., and Seierstad, I. A.: The Norwegian Earth System Model, NorESM1-M – Part 2: Climate response and scenario projections, Geosci. Model Dev., 6, 389–415, 2013.

---

## Author Comment (AC3) · 29 Nov 2019

The comment was uploaded in the form of a supplement

Please also note the supplement to this comment:
https://www.clim-past-discuss.net/cp-2019-102/cp-2019-102-AC3-supplement.pdf

———————————————

---

## Author Comment (AC4) · 29 Nov 2019

Point to point response to reviewers' comments

*The comments are in blue, and our responses are in black.*

Response to Reviewer 2:

Comment from reviewer 2
This manuscript presents initial results from the Nor-ESM modeling group for two simulations of the mid-Pliocene Warm Period experiment (Eoi400 of Haywood et al., 2016b), the core paleoclimate experiment of the Pliocene Model Intercomparison Project (PlioMIP) Phase 2, as a contribution to CMIP6. The simulations were run with the older NorESM-L (also used in PlioMIP1) and the more recent NorESM1-F. Six key diagnostic variables were examined, and the NorESM-L PlioMIP2 run was also compared to the group's earlier efforts for PlioMIP Phase 1. This is a solid contribution to the PlioMIP2 effort, and I recommend publication of this paper, subject to minor modifications to address the comments raised here.

We thank the reviewer for the positive assessment and constructive comments on our manuscript. We respond to the reviewer's comments below.

General comments:
Of note is the authors' finding that the NorESM1-F mid-Pliocene simulation actually warms less (+1.7 degC global mean SAT, +1.2 degC global mean annual SST compared to PI control) than the equivalent simulation with the older NorESM-L model (+2.1degC global mean SAT, +1.5 degC global mean annual SST). This relative cooling of higher resolution model compared to the lower resolution model is not entirely expected, nor is both model versions' relative cooling of the Pliocene simulation compared to PI control; it is also not consistent with some of the other PlioMIP2 experiments already reported (MRI-CGCM2.3, CCSM4, IPSL-CM5A). Furthermore, the NorESM1-F Eoi400 simulation is itself cooler than the equivalent PlioMIP1 simulation (-1.1 degC global mean SAT). The authors attribute this primarily to the change in paleogeographic boundary conditions from PlioMIP1 to PlioMIP2. While this is certainly possible for NorESM-L, paleogeography alone cannot address why the newer NorESM1-F is generally not as warm as the older NorESM-L under PlioMIP2 boundary conditions.
It would be helpful to know how the equilibrium climate sensitivity differs between the two model versions. It would also be useful to know whether the authors have previously documented any differences in run results owing the horizontal grid resolution differences between the coarser grid NorESM-L and finer grid NorESM1-F.

NorESM-L is the low resolution version of the NorESM1-M (the CMIP5 version of the NorESM) and is designed for simulations of past climates (Zhang et al., 2012;

Bents et al., 2013). NorESM1-F is different from NorESM1-M with new implementations and code developments, including some updates to the ocean physics and modification in the atmosphere component, and aiming to have a similar performance with adequate resolution, process representations, and improved integration efficiency (Guo et al., 2019).

The estimated equilibrium climate sensitivity of NorESM1-F is 2.29 ℃ (Guo et al., 2019), which is lower than that of NorESM-L (3.1 ℃, Haywood et al., 2013). Compared to NorESM-L, the estimated lower equilibrium climate sensitivity of NorESM1-F may, at least partly, explain its simulated lower warming in the Pliocene.

NorESM-L is the low resolution version of NorESM1-M, and the latter has the same resolution as NorESM1-F. With the same resolution, the estimated equilibrium climate sensitivity of NorESM1-F is still lower than NorESM1-M (2.29 ℃ *vs.* 2.9 ℃) (Iversen et al., 2013; Guo et al., 2019). Therefore, the resolution difference between the two versions seems not to be the most important reason for the simulated lower warming in Pliocene with NorESM1-F as compared to NorESM-L. It is difficult to make a 'clean' comparison in terms of resolution only, as there are also significant code changes in the physics between the two models.

Specific comments:

Page 3, lines 17-19 – Can the authors be more specific about the additional improvements to the MICOM ocean component of NorESM? Also, the authors note that NorESM1-F was run without the CAM4-Oslo advanced scheme for interactions between aerosols and clouds. Is there a CMIP6 PI control run available for NorESM1-F with the CAM4-Oslo scheme enabled, to compare with the NorESM1-F described here? I wonder whether the absence of this scheme with the newer model might also contribute to the lower magnitude of warming in the mid-Pliocene run described here.

Revised. We added more details about NorESM1-F in Section 2.2 in the revised manuscript.

Compared to NorESM1-M, there are some updates in the ocean physics in NorESM1-F. NorESM1-F employs a method to reduce sea ice thickness biases in shelf regions and modifies the methods of parameterization of oceanic mesoscale eddies and the vertical mixing (Guo et al., 2019). With those updates to the ocean physics, NorESM1-F provides reasonable simulations of sea ice and AMOC (Guo et al., 2019).

There is no CMIP6 PI run available for NorESM1-F with the CAM4-Oslo scheme enabled. To limit model complexity and speed up model integration, both NorESM-L and NorESM1-F use the standard, prescribed aerosol chemistry of CAM4 rather than that of CAM4-Oslo. We have clarified this point in Section 2 of model descriptions.

Page 4, lines 10-15 – It is unclear why NorESM-L was run with different PI greenhouse gas values (280 ppmv, 270 ppbv, 760 ppbv of CO2, N2O and CH4 respectively) compared to NorESM1-F (284.7 ppmv, 275.68 ppbv, 791.6 ppbv of CO2, N2O and CH4 respectively) for the PI control run. If this was done so that the only key difference between PlioMIP1 and PlioMIP2 simulations with NorESM-L was the new paleogeographic reconstruction for PlioMIP2, it would be helpful to **clarify that**. If, however, the code for NorESM-L has been updated since 2012, it would be necessary to provide more detail on what has changed.

There is no change in the code of NorESM-L since 2012.

The choice of PI greenhouse gas vales with NorESM-L is based on the guideline of PMIP. However, the choice of PI greenhouse gas vales with NorESM1-F is based on the CMIP5 guideline.

Section 4.5 Sea Ice, pages 6-7 – The sea ice differences between NorESM-L and NorESM1-F merit some additional discussion, especially for the Southern Ocean around Antarctica. Can the authors elaborate on why NorESM1-F is producing so much more ice in this region?

We re-organized some sentences in Section 5.1 to give some explanation. Simulated Pliocene southward ocean heat transport to the Southern Ocean is reduced according to NorESM1-F, but increased according to NorESM-L (Fig. 7), which partly explains the reduction in the Southern Ocean sea ice extent being more pronounced for NorESM-L than it is according to NorESM1-F (Fig. 5) (Page 9, Line 8–11 in the manuscript with the revisions marked).

The divergent responses in sea ice are more likely to be associated with the Southern Ocean stratification in Pliocene simulated between the two versions. NorESM-L simulates increased ventilation in the Southern Ocean, while NorESM1-F does not. The Pliocene sea ice reduction is larger in NorESM-L than in NorESM1-F. However, it remains difficult to fully explain the divergent responses. We pointed out the possible reason in the discussion: "Such divergent responses in Southern Ocean stratifications also appeared in the PlioMIP1 simulations (Zhang, Z. et al., 2013a). It remains difficult to fully explain the divergent responses. The explanation is likely related to the updated ocean physics and/or higher resolution in NorESM1-F, when compared to NorESM-L" (Page 9, Line 25–28 in the manuscript with the revisions marked).

Technical comments:

Page 2, line 21 – There is a reference here to Zhang, R. et al 2013. Elsewhere, there are cites for Zhang, R. et al 2013a and Zhang, R. et al 2013b, but there are a total of three Zhang, R. et al 2013 references in the reference list. These should be renumbered to avoid confusion.

In the manuscript, there is only one reference to Zhang, R. et al 2013. And the

other two references are to Zhang, Z. et al. 2013a and Zhang, Z. et al. 2013b.

Page 4, line 4 – How many ocean layers for NorESM1-F?

There are 53 vertical layers in the ocean component of NorESM1-F.

We added this information in Section 2.2. Please see Page 4, Line 13 in the manuscript with the revisions marked.

Page 5, lines 15-17 – The description of the regional temperature highs is not consistent with Table 3 – perhaps because Table 3 lists regions from SH pole to NH pole, which is a little non-intuitive.

Revised.

In both tables 3 and 5, the order of the region list was not right in the manuscript. After we corrected this in the revised version, the listing regions start from the NH and are consistent with the description in the context.

Please see tables 3 and 5.

Page 5, line 21 – Should read "circum-Arctic" rather than "circus-Arctic"

Revised.

Page 9, lines 6-7 – Perhaps say "In contrast," rather than "On the contrary,"

Revised.

Table 3 and Table 5 – listing regions starting with the NH polar region at the top would be a more intuitive way to present this information

Revised.

Referrences

Bentsen, M., Bethke, I., Debernard, J.B., Iversen, T., Kirkevåg, A., Seland, Ø., Drange, H., Roelandt, C., Seierstad, I.A., Hoose, C., and Kristjánsson, J.E.: The Norwegian Earth System Model, NorESM1-M – Part 1: Description and basic evaluation of the physical climate. Geosci. Model Dev. 6, 687–720, 2013.

Guo, C., Bentsen, M., Bethke, I., Ilicak, M., Tjiputra, J., Toniazzo, T., Schwinger, J., and Otterå, O.H.: Description and evaluation of NorESM1-F: a fast version of the Norwegian Earth System Model (NorESM). Geosci. Model Dev. 12, 343–362, 2019.

Haywood, A.M., Hill, D.J., Dolan, A.M., Otto-Bliesner, B.L., Bragg, F., Chan, W.L., Chandler, M.A., Contoux, C., Dowsett, H.J., Jost, A., Kamae, Y., Lohmann, G., Lunt, D.J., Abe-Ouchi, A., Pickering, S.J., Ramstein, G., Rosenbloom, N.A., Salzmann, U., Sohl, L., Stepanek, C., Ueda, H., Yan, Q., and Zhang, Z.: Large-scale features of Pliocene climate: results from the Pliocene Model Intercomparison Project. Clim. Past 9, 191–209, 2013.

Zhang, Z.S., Nisancioglu, K., Bentsen, M., Tjiputra, J., Bethke, I., Yan, Q., Risebrobakken, B., Andersson, C., and Jansen, E.: Pre-industrial and mid-Pliocene simulations with NorESM-L. Geosci. Model Dev. 5, 523−533,

2012.

Zhang, Z.S., Nisancioglu, K.H., Chandler, M.A., Haywood, A.M., Otto-Bliesner, B.L., Ramstein, G., Stepanek, C., Abe-Ouchi, A., Chan, W.L., Bragg, F.J., Contoux, C., Dolan, A.M., Hill, D.J., Jost, A., Kamae, Y., Lohmann, G., Lunt, D.J., Rosenbloom, N.A., Sohl, L.E., and Ueda, H.: Mid-pliocene Atlantic Meridional Overturning Circulation not unlike modern. Clim. Past 9, 14955–11504, 2013.